# A hyperconformal dual-modal metaskin for well-defined and high-precision contextual interactions

Shifan Yu[1], Zhenzhou Ji[1], Lei Liu[1], Zijian Huang[1], Yanhao Luo[1], Huasen Wang[1], Ruize Wangyuan[1], Ziquan Guo[1], Zhong Chen [1], Qingliang Liao[2,3], Yuanjin Zheng[4] & Xinqin Liao [1] ✉

Proprioception and touch serve as complementary sensory modalities to coordinate hand kinematics and recognize users' intent for precise interactions. However, current motion-tracking electronics remain bulky and insufficiently precise. Accurately decoding both is also challenging owing to the mechanical crosstalk of endogenous and exogenous deformations. Here, we report a hyperconformal dual-modal (HDM) metaskin for interactive hand motion interpretation. The metaskin integrates a strongly coupled hydrophilic interface with a two-step transfer strategy to minimize interfacial mechanical losses. The 10-µm-scale hyperconformal film is highly sensitive to intricate skin stretches while minimizing signal distortion. It accurately tracks skin stretches as well as touch locations and translates them into polar signals, which are individually salient. This approach enables a differentiable signaling topology within one single data channel without burdening structural complexity to the metaskin. When combined with temporal differential calculations and time-series machine learning network, the metaskin extracts interactive context and action cues from the low-dimensional data. This phenomenon is further exemplified through demonstrations in contextual navigation, typing and control integration, and multi-scenario object interaction. We demonstrate this fundamental approach in advanced skin-integrated electronics, highlighting its potential for instinctive interaction paradigms and paving the way for augmented somatosensation recognition.

The human somatosensory system encompasses a dynamic interplay between proprioception and touch, which underpins individual proximal cognition[1,2]. Proprioception allows for humans to track our limb posture and movements, and the sense of touch, as an exteroception, provides percept into the attributes of the objects we contact[3,4]. The integration of these two complementary sensations enables the human body to go beyond mere reflexive reactions, empowering humans with the essential capacity to adaptively modify their behaviors in response to ever changing environments[5]. For example, preschool children learn to stack building blocks of different shapes and orientations by modulating their hand posture and applying appropriate force. These two sensory modalities are not

[1]Department of Electronic Science, Xiamen University, Xiamen, China. [2]Academy for Advanced Interdisciplinary Science and Technology, Key Laboratory of Advanced Materials and Devices for Post-Moore Chips Ministry of Education, University of Science and Technology Beijing, Beijing, China. [3]Beijing Key Laboratory for Advanced Energy Materials and Technologies, School of Materials Science and Engineering, University of Science and Technology Beijing, Beijing, China. [4]School of Electrical and Electronic Engineering, Nanyang Technological University, Singapore, Singapore. ✉e-mail: liaoxinqin@xmu.edu.cn

discrete within the continuous temporal framework, as one modality serving as a prior context influences the decision-making and experience of the other[6]. To temporally integrate both perceptual modes, a concept named "bodily context" is introduced, which associates proprioceptive hand states with touch patterns. This approach not only decouples touch stimuli from hand motions but also contextualizes them: identical tactile inputs yield distinct perceptual results under different postures, and vice versa. By resolving these multimodal relationships, the same touch action can express divergent intentions on the basis of its context, enabling natural motion-chain interpretation. The motion coupling based on the bodily context will revolutionize the interactive model, enabling cognition and judgment depending on the context in which it occurs.

Wearable devices capture and interpret hand motion and touch to convey the interactive intentions of humans, such as object manipulation or gesture recognition[7–11]. Gesture recognition complements perception beyond vision, playing an indispensable role in immersive virtual reality and communication assistance. Recent advances in markerless gesture measurement have aimed to achieve a natural description of hand movements[12–14]. However, touch information is often overlooked in gesture recognition frameworks, resulting in systems that still deviate from the natural kinematic chain of hand actions. By integrating contextual awareness with motion cognition, the hand's proprioceptive state, i.e., the bodily context, directly influences touch perception, and vice versa. This integration breaks the traditional occlusion between proprioceptive motion and tactile modalities.

Most wearable devices proposed for data collection require bulky portable carriers, such as haptic gloves, which cause intrinsic mechanical losses and poor precision[15]. Thus, imperceptible soft e-skins that capture high-precision contact information are gaining attention, driven by a strong desire to produce devices that can be integrated into everyday actions without constraining the users[16]. Recent advances in the in-situ printing of nanomesh e-skin have achieved biomimetic sensing and imperceptible implementation, directly mapping microscale skin stretches to proprioception[17]. However, these devices often present discrete or single modal haptic cues. They are tailored to predefined procedure settings, and lack a focus on cross-scenario generalization capabilities and adaptive performance. Therefore, exploring the dependence between proprioception and touch helps to offer new insight into the adjustment of interaction feedback and manipulation mode switching on the basis of the bodily context. This is especially important when trying to replicate or enhance human-like actions in machines or interfaces. However, owing to the similarity between skin stretch and touch deformation signals, artificial proprioceptors and touch sensing arrays suffer mechanical crosstalk[18]. This explains why touch sensing arrays employ serpentine interconnects to counteract interference from stretching signals[19], or why artificial proprioceptors target negligible pressure sensitivity[20,21]. Additionally, they typically require multichannel data acquisition to isolate multiple joints and sites, ultimately resulting in higher-than-expected complexity and spatiotemporal misalignment[22]. To address the existing research gap, particularly the challenge of effectively translating and separating complementary somatosensory information, it is essential to develop a well-defined signaling strategy and ensure its accessible implementation.

In this article, we report a hyperconformal dual-modal (HDM) metaskin. The metaskin leverages the integration of the hydrophilic thin substrate (8–10 μm) and highly compatible water-based Ag nanocomposite sensitive path, minimizing impediments to hand movements. A two-step transfer strategy exploits tunable maintenance and separation of hydrophilic films on hydrophobic interfaces, facilitating seamless, strain-free integration onto any part of the skin. Benefiting from the nanocomposite conductive network of sensitive paths, metaskin demonstrates high strain sensitivity (gauge factor: 32.45 from 0 to 20%) and robust electromechanical stability, while remaining insensitive to humidity and temperature. It can capture and interpret subtle resistance changes caused by skin stretching, while also pinpointing relative locations of electrical contact. Even under stretched conditions, touch positions can be precisely localized through decoupling computations. The HDM metaskin can differentiate proprioceptive and touch actions into well-defined signals from single-channel data. This capability is attributed to the polarity difference between strain signals and touch signals, that is, stretch-induced resistance changes representing proprioceptive strain consistently increase, whereas resistance changes associated with touch positioning proportionally decrease. On the basis of a dual interpretation of somatic motions, different hand movements can be regarded as combinations of bodily contextual signals and event–action signals. By seamlessly integrating proprioceptive and exteroceptive data, this approach enhances the coupling of contextual and kinematic information, enabling more precise intention recognition and task adaptation. These features facilitate temporal differentiation to focus on polarity detection, time intervals, and amplitude quantification within sequential signals, achieving feasible in-sensor calculations. We demonstrate that the metaskin exhibits exceptional adaptivity in extensive contextual applications, such as switching postures while navigation, and operation integration of virtual typing and mouse control. Additionally, touch addressing can be used to describe object contact postures, providing a high-fidelity interpretation of the body context in object interaction applications, thereby enhancing scene generalization and interaction intent correction in time-series machine learning models. We believe these demonstrations will inspire broader domains, including soft robotics and embodied interactions, paving the way for future advancements in somatosensory-based systems.

## Results
### Mechanism of somatosensory proprioception and exteroception

Proprioceptive and tactile sensing rely on mechanotransduction, the process of encoding endogenous and exogenous mechanical deformations of tissues into neural signals[1,23]. Cutaneous mechanoreceptors, located within the skin, generate signals related to contact distribution through touch receptors; whereas proprioceptors, embedded in muscles, tendons, and ligaments, convey information about limb positions and somatic motions[24]. The signals from different afferent fiber groups collectively depict a neural representation of the ambient objects (Fig. 1a). Within the framework of intelligent interaction, these two sensations provide crucial complementary information reflecting the embodied agent and the environment, and serve as a foundational prior for adaptation to the unstructured environment.

Artificial devices for proprioception and exteroception typically employ distributed multilayer structures. However, tactile sensors typically require densely packed sensing arrays to ensure spatial discrimination, which results in highly localized data generation. Additionally, joint bending introduces an undesired normal force component to tactile sensors, resulting in unresolvable signal differentiation issues[25,26]. Thus, the obstacle to achieving signal duality in dual-modal sensing systems lies in the isotropy of sensitive variables and undecipherable signal masking. Furthermore, multilayered sensors lack conformal integration across crinkled skin, causing unavoidable sensation disturbance to tiny movements[27]. In contrast, our fabricated HDM metaskin features an in-plane structure that closely follows the topography of the skin, enabling the separation of strain and contact sensing. The metaskin compresses longitudinal deformation and closely conforms to the skin's natural folds, ensuring high precision and an imperceptible user experience. Importantly, continuous and common mode data eliminate delays across multiple channels during data integration, enabling the subdivision of interactive intentions in diverse bodily contexts[28].

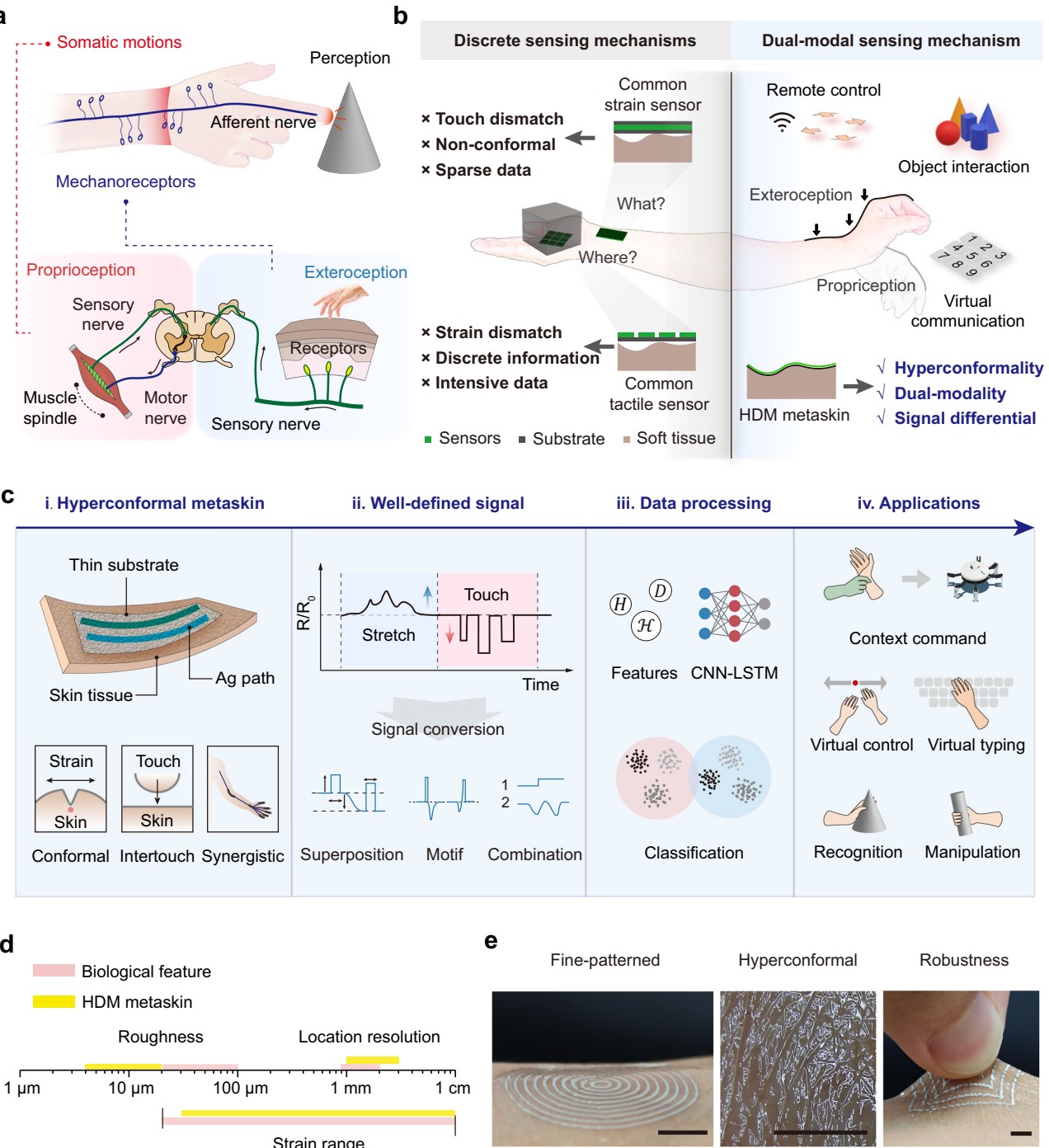

**Fig. 1 | The HDM metaskin analogous to human sensory functions. a** Illustration of somatosensory mechanisms, differentiating between proprioception and exteroception through nerve pathways. Contains photos by jannoon028 and littlestocker via Freepik License. **b** Integration of decoupled perception mechanisms into a dual-modal, tissue-conformal sensor network, distinguishing between "strain sensing" and "tactile sensing" for precise command, recognition, and communication. Contains photos by kwangmoop and tehchesiong via Freepik License. **c** Device, mechanism, and data processing methods of HDM metaskin for multiple applications. i. HDM meta-skin on skin, featuring ultra-thin, conformal substrate and exposed interaction path. ii. Signal difference under stretch and touch modalities. iii. Data processing through data fusion and machine learning algorithms. iv. Applications include contextual command, virtual control and typing, and object interaction. Contains icons by macrovector via Freepik License. **d** Comparative scale between biological skin and HDM meta-skin. **e** Demonstration of sensor properties of the HDM metaskin on skin. Scale bar: 5 mm.

Figure 1c shows an overview of our proposed HDM metaskin for contextual interactions. It consists solely of a film substrate and strongly bonded nanocomposite conductive paths, conformally adhering to the skin surface. This configuration can respond to subtle skin strains and conductive intertouch, achieving synergistic perception of bodily motions. The complementary nature of stretch and touch signals, characterized by their opposing modalities, enables precise differentiation and supports signal superposition, motifs, and complex combinations. By integrating decoupling algorithms and temporal learning models, this collaborative bodily cognition can be applied to interpret diverse forms of hand movements and support contextual interactions. As a skin-extension technology, the metaskin

is engineered to emulate key biological features, such as surface roughness, spatial resolution, and strain adaptability, ensuring compatibility with natural human tactile functions[29,30] (Fig. 1d). These advanced capabilities are realized through robustly bonded soft functional materials, which allow for in-plane pattern extensibility, seamless conformal integration with the skin, and exceptional mechanical durability under dynamic conditions (Fig. 1e).

## Preparation and installation of the HDM metaskin

To construct conformal and stretchable devices on the skin, any thermal or mechanical predeformation can adversely affect the final test results. To address this, we implemented the following strategies: (1) A water-based Ag nanocomposite paste with mild curing conditions was adopted, which was composed of Ag nanoparticles, Ag nanowires, and a waterborne polyurethane (WPU) binder (Fig. 2a). Compared with conventional organic silver pastes that require sintering temperatures as high as 130 °C, water-based pastes cure at a significantly lower temperature of only 60 °C. Figure S1 shows the thermally induced deformation of the substrate film. High temperatures can cause severe distortion of the substrate membrane during the device fabrication stage. (2) The substrate film employed a hydrophilic material to ensure strong interfacial interactions with the water-based Ag paste. (3) A two-step transfer strategy was utilized to ensure robust interfacial anchoring of the Ag nanocomposite and prevent local distortions throughout the skin-transfer process.

A dispensing printing technique was employed to achieve customized pattern adaptation of conductive paths across a range of profiles. The Ag nanocomposite paste was printed onto a WPU film mounted on a release substrate. Owing to the hydrogen bonding interactions among the water-based binders, the Ag nanocomposite paths can firmly adhere to the WPU film, outweighing the typical van der Waals interactions (Fig. 2a, b and Fig. S1). Because the thin-film substrate was susceptible to distortion without a support film and because the conductive paths needed to be the outer surface, a strainless transfer strategy was proposed, which is appropriate for any irregular target surface. Although direct transfer to the skin could theoretically be achieved by printing the Ag path first, the printed patterns interfered with the uniform distribution of the spin-coated film, leading to poorer interfacial bonding. By exploiting the differences in surface energy and interfacial roughness, the HDM metaskin, initially supported on release paper, can be readily transferred onto a softer silicone film through a simple, pressure-assisted method. The second inversion step employed a pressure-sensitive adhesive (PSA) to affix the HDM metaskin onto the target surface, ensuring stable bonding (a detailed transfer process is shown in Fig. S2). The metaskin can conform seamlessly to intricate leaf veins and skin wrinkles, demonstrating excellent adaptability to uneven surfaces without distortion in linear conductive paths (Fig. S3).

We compared the interfacial adhesion strength of composite silver patterns printed on different substrates by observing the extent of damage after repeated 3 M tape peeling tests (Fig. 2b and Fig. S4). The water-based silver paste exhibited poor abrasion resistance on hydrophobic films. However, when printed on the polyurethane substrate, the silver patterns withstood over 100 peeling cycles, and with further water-based modifications, the silver patterns experienced only minimal damage even after 200 peeling cycles. Notably, the resistance of the printed paths should fall within the ideal range of hundreds of ohms to ensure measurable variations, which could be achieved by regulating the printing parameters (Fig. S5).

Next, we focused on interface engineering to elucidate the issues involved in the formation and transfer of thin films. Owing to the hydrophilic nature of WPU, thin films cannot be successfully spin-coated from dilute solutions on hydrophobic substrates[31]. As shown in Fig. 2c, when a 30 vol% WPU solution was spin-coated at 1000 r·min⁻¹, the liquid underwent annular contraction driven by the imbalance of

Young–Laplace forces. This phenomenon occurred because dilute solutions tend to exhibit higher surface tension due to water aggregation, coupled with low viscous resistance, which promotes liquid redistribution. According to Young's equation:

$$\cos\theta = \frac{\gamma_{SG} - \gamma_{SL}}{\gamma_{LG}} \qquad (1)$$

where $\theta$ is the contact angle between the liquid and the solid, and where $\gamma_{SG}$, $\gamma_{SL}$, and $\gamma_{LG}$ represent the interfacial energies of the solid–gas, solid–liquid, and liquid–gas interfaces, respectively. After spin-coating, the contact angle on the hydrophobic surface tends to be large. At this point, the liquid surface underwent spontaneous contraction. In contrast, at 50 vol% and 3000 r·min⁻¹, the film spread evenly because of the reduced surface tension of the higher concentration solution, combined with a stronger centrifugal force, overcoming retraction forces. Under these conditions, the resulting film thickness was 8–10 μm. As the spin speed increased, the film thickness progressively decreased, ultimately reaching 4 μm at 4000 r·min⁻¹ (Fig. S6). Mechanical testing demonstrated that the thin-film substrate exhibited negligible tensile forces on the skin, minimizing any foreign body sensation (Fig. S7).

The transfer process involves the interaction and migration of four key interfaces: the release paper, the WPU, the silicone transfer film, and the target interface. In this context, each transfer inherently replaces a low-interaction interface with a high-interaction interface, placing additional demands on the transfer auxiliary film to establish a transient adsorption state[32]. Excessively strong van der Waals forces at the interface can impede the peeling process of the second transfer, whereas an uneven distribution of van der Waals forces may lead to localized strain, compromising the uniformity of the WPU film (Fig. 2d). Accordingly, three critical requirements for the auxiliary transfer film arise: it must exhibit low inherent surface energy, form a sufficiently strong van der Waals interactions with the WPU film, and ensure uniform van der Waals adsorption. We observed that despite the silicone film having lower interface energy, its smooth surface facilitates stable van der Waals adsorption with the WPU. During the first transfer, this adsorption force is notably stronger than the interaction between the slightly rougher release paper and the WPU, thereby ensuring strain-free peeling. During the second transfer step, PSA ensured the strong adhesion of the WPU film and easy separation from the silicone substrate (as shown in Fig. S8 and Supplementary Movie 1). Notably, this outcome is not merely a result of surface energy ranking, even though the silicone film typically has a lower surface energy than does the release paper. Owing to the strong adhesion between the nanocomposite paste and the WPU substrate, the transfer of HDM metaskin transferred to the skin exhibits exceptional electrical and mechanical robustness. It maintained stable electrical resistance under prolonged frictional wear (30 m, 2 N, 10 cm s⁻¹) and high humidity (90% RH), while resisting resistance changes under thermal conditions. Even across the safe temperature range (25–60 °C), the resistance variation remained below 5% (Fig. 2e and Fig. S9). These attributes ensure reliable performance in diverse and dynamic environments.

## Combination and decoupling of the HDM metaskin

The integrated design of the HDM metaskin leverages the seamless integration of mechanical stretch with circuit principles, enabling independent contributions to proprioception and touch sensing. It is composed of two functional components, parallel conductive pathways and additional electrode triggers, which can be placed on the fingertip or the interactive object (Fig. 3a). For proprioceptive sensing, the linear conductive pathways convert uniaxial skin strain into resistance changes, governed by microcrack expansion within the Ag

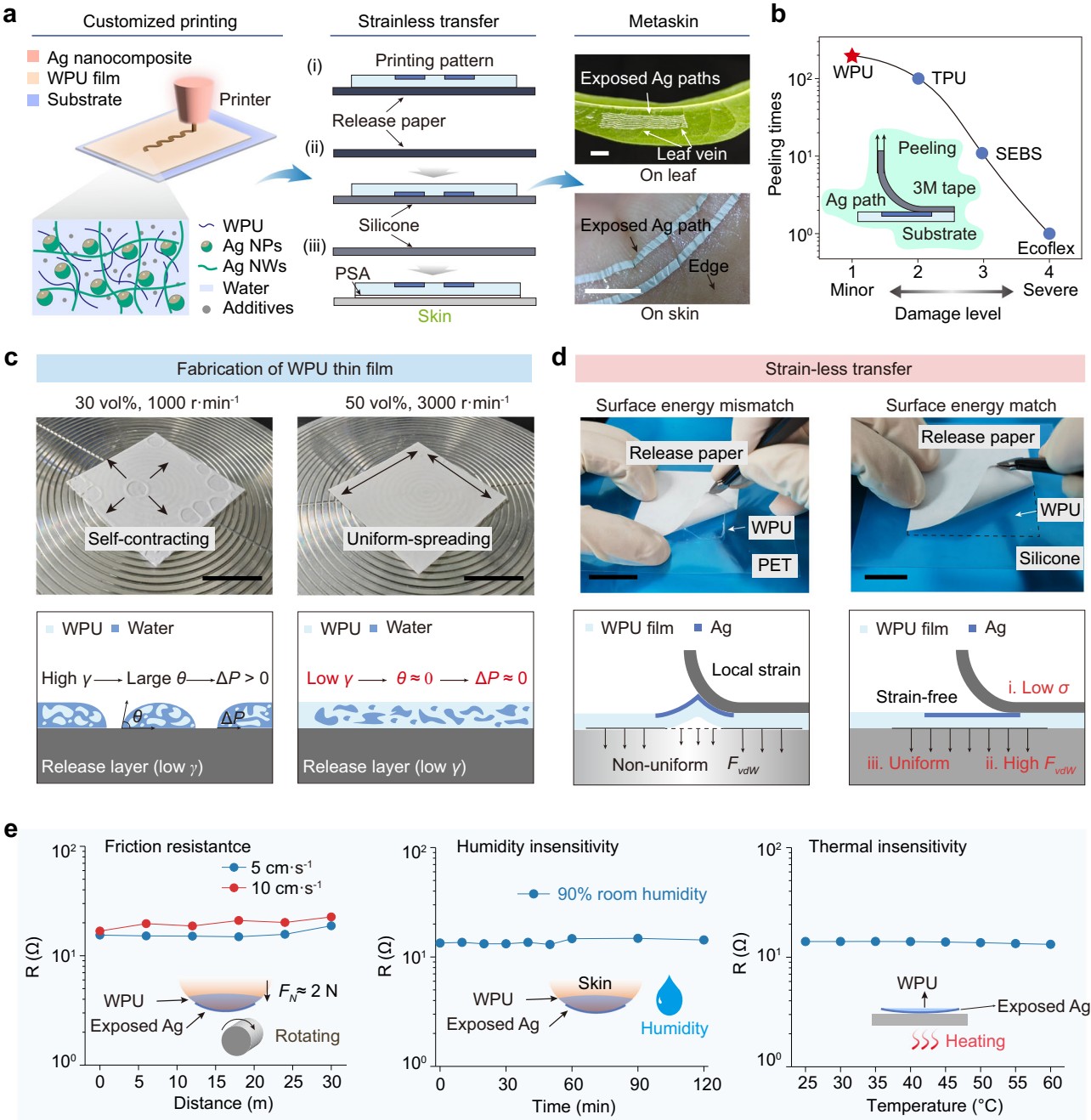

**Fig. 2 | Fabrication and transfer of the HDM meta-skin. a** Printing, transfer, and conformal performance of the HDM metaskin. Through a strain-free transfer strategy, customized printed patterns are effectively. transferred onto irregular surfaces, demonstrating intimate conformability to both a leaf and human skin. Scale bar: 1 cm. (*WPU* waterborne polyurethane; *Ag NPs* silver nanoparticles; *Ag NWs* silver nanowires; *PSA* pressure-sensitive adhesive) (**b**) Adhesion durability of printed Ag paths on different substrates under. cyclic peeling tests. The Ag paths on WPU withstand the highest number of peeling cycles while showing. minimal damage. (*WPU* waterborne polyurethane; *TPU* thermoplastic urethanes; *SEBS* styrene-ethylenebutene- styrene block copolymer) (**c**) Surface mechanisms the

spin coating of the WPU film. Scale bar: 2 cm. (γ represents the surface tension, θ represents the contact angle and ΔP represents the pressure differential. between the exterior and interior of the solution) (**d**) Detachment of WPU and strain-less transfer to the. transition layer. Compared to polyethylene terephthalate (PET) surface, silicone exhibits a more uniform. surface energy match with WPU and stronger adhesion to the release paper. Scale bar: 2 cm. (*FvdW* represents. Van der Waals forces at the interface) (**e**) Robust performance of the HDM metaskin across friction (*FN*≈2 N), humidity conditions, and temperature ranges from 25 °C to 60 °C.

nanocomposite network. The scanning electronic microscope (SEM) images revealed that the Ag nanoparticles (NPs) and Ag nanowires (NWs) collectively formed interlocking networks, moreover, the WPU binder facilitated the formation of an interpenetrating interface (Fig. S10). This delamination-free interface effectively reduces mechanical transmission losses in the normal direction, thereby

enabling the sensor pathways to stretch in synchrony with the deformation of the skin[33]. Upon stretching, Ag nanoaggregates form microcracks, inducing resistance changes proportional to the applied strain. The embedded Ag NW network, which serves as an anchoring phase, ensures resistance stability under prolonged strain[34–36] (Fig. S11). For touch sensing, the primary sensor works synergistically

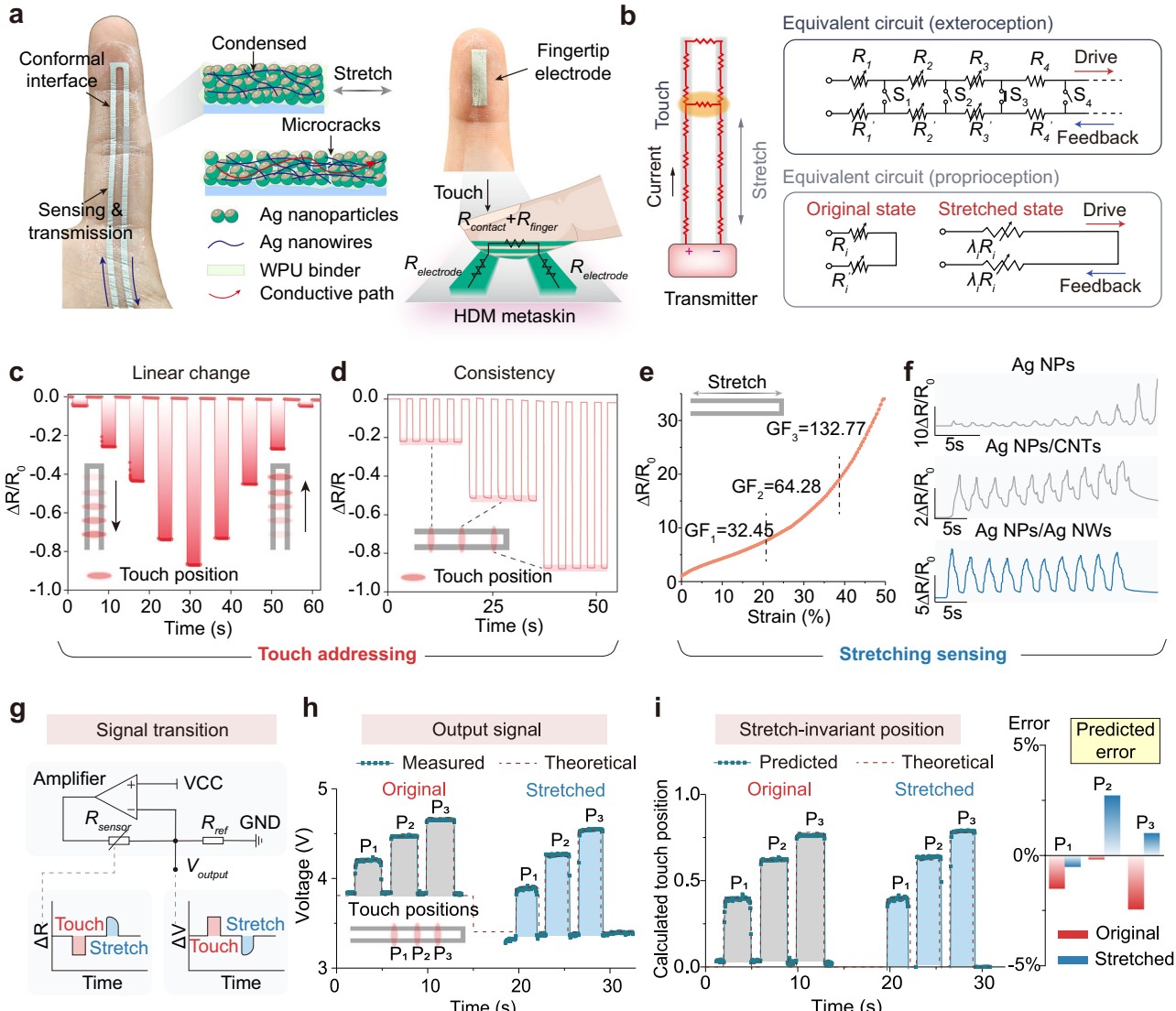

**Fig. 3 | Well-defined signals of strain sensing and touch positioning sensing.** **a** Photograph of the HDM. metaskin conformally attached to the fingers, including a parallel conductive pathway and a touch trigger. Insets show microscopic mechanisms of strain sensing and touch localization functions. ($R_{finger}$, $R_{electrode}$ and $R_{contact}$ correspond to the finger electrode resistance, sensor electrode resistance, and contact resistance, respectively.) **b** Equivalent circuit for exteroceptive and proprioceptive sensing modes. ($R$ and $R'$ represent. the equivalent resistances. $\lambda$ represents the stretching coefficient.) **c** Linear resistance changes as touch positions on the meta-skin are progressively varied. Touch resistance directly reflects the touch position, with resistance changes always being negative. **d** Consistent signal output during repeated touches at same locations. **e** Stretch response curve

across a range of 0% to 50% strain. The segment gauge factors GF1, GF2, and GF3 were 32.45, 64.28, and 132.77, respectively. **f** Comparative stretch responses at 5% strain. were evaluated for sensors fabricated with Ag NP, Ag NP/CNT, and Ag NW pastes. The Ag NP/Ag. NW nanocomposite sensor demonstrated the most stable performance under cyclic loading conditions. **g** Voltage divider circuit diagram and corresponding signal patterns, demonstrating the conversion process. from resistance to voltage. ($R_{sensor}$ and $R_{ref}$ represent the sensor resistance and the reference resistance, respectively.) **h**, **i** Decoupling calculation of touch positions under stretching conditions, comparing. measured results to theoretical values for validation. (P1, P2 and P3 means three different touch positions).

with fingertip-mounted triggers to achieve touch localization. Since the contact resistance upon contact is negligible (Fig. S12), the overall resistance was determined solely by the closed circuit between the contact point and the terminal, which was proportional to the initial resistance. In other words, we established a dedicated mode for sensing the position of the stimulus, since the system only responded when contacted with external trigger electrodes. One key benefit of this approach is that it keeps the triggers easy to install, while still ensuring specificity in interactive events and ease of integration into various applications. The dual-modality functionality is further clarified in the detailed circuit diagram (Fig. 3b), which illustrates the interaction between stretch-induced and contact-induced signals. The response resistance when stretching and touching can be expressed

as:

$$R_s = \sum_{i=1}^{n} (\lambda_i R_i + \lambda_i R_i')(\lambda_i \geq 1) \qquad (2)$$

$$R_t = \sum_{i=1}^{p} (\lambda_i R_i + \lambda_i R_i')(\lambda_i \geq 1, p < n) \qquad (3)$$

where, $R_i$ represents the resistance of the segmental resistor units and where $\lambda_i$ denotes the stretching factor. $R_s$ and $R_t$ correspond to the resistance upon stretching and touching, respectively. Evidently, $R_s$ remained consistently higher than the initial resistance as the HDM

metaskin was stretched. In contrast, $R_t$ decreased proportionally to the total resistance and showed no dependence on the degree of stretching. Thus, the two-terminal device operated through a single circuit loop, where the intrinsic resistance increased during stretching, whereas the electric contact mechanism induced linear reductions on the basis of the total loop resistance.

The antithetic response tendency generates well-defined signals without a complex algorithm. To validate this, the relative resistance changes were measured under two activation modes. A negative resistance change was observed when contact stimulation was applied with a conductive stick every 2.5 cm alongside the active path (Fig. 3c). Here, owing to the direct electrical contact of the conductive path with the trigger, the resistance signal immediately responded without precompression delay or mechanical fatigue. The consistency of the signal at the same contact position was also ensured (Fig. 3d). In contrast, the stretch-induced resistance signal was positive and initiated from the base value. Owing to the interlocking network and interpenetrating interfaces, the Ag nanocomposite path exhibited a highly sensitive response, with a gauge factor of 32.45 in the strain range of 0 to 20%, and demonstrated enhanced sensitivity up to a strain of 50% (Fig. 3e). Additionally, we compared the cyclic resistance recoveries of Ag NPs, Ag NPs/carbon nanotubes (CNTs), and Ag NPs/Ag NWs to highlight the crucial role of the composite network (Fig. 3f). Without the anchoring effect of Ag NWs, the Ag NP path experienced pronounced signal fluctuations and drift under a cyclic strain of 5%. This was probably attributed to the unstable expansion and delayed recovery of the nanocracks. Introducing CNTs partially reduced these fluctuations, yet minor variations remained. In contrast, incorporating Ag NWs resulted in a more robust conductive network, yielding consistently lower and more stable resistance changes.

With the goal of mathematically decoupling overlay response patterns, a voltage divider module was implemented for signal acquisition. This module transformed the initial resistance into a reciprocal voltage output (Fig. 3g), ensuring that the resulting output voltage was constrained within the range of 0-5 V. By maintaining a fixed voltage interval, stretch and touch variations could be decoupled via proportional scaling. The output voltage exhibited baseline-referenced bipolar variation, in which positive shifts corresponded to touch events, and negative shifts represented stretching. The final touch position was determined by the output voltage measured across the reference resistor, with the detailed calculation method provided in Note S1. This design facilitated efficient signal processing, while maintaining compatibility with the operating standards of portable measurement systems (limit of 5 V). As shown in Fig. S13, the output voltage was influenced by both the touch position and the stretch factor. However, the decoupling algorithm successfully resolved the relative touch position, demonstrating robust performance under varying stretch conditions. To validate the decoupling accuracy, a stretch–touch experiment was performed alongside a corresponding time-series simulation. Three points on the HDM metaskin ($P_1$, $P_2$, and $P_3$ as shown in Fig. 3h) were randomly selected, and subjected to touch stimulation under both a relaxed state and a stretched state. Thus, each touch point generated two directly measured voltages. The corresponding simulated voltages were derived theoretically from real-time resistance variations during relaxation, stretching, and touching. The comparison between the experimental and simulated results demonstrated a high degree of consistency in predicting relative touch positions (Fig. 3i), with a maximum prediction error of less than 2.6%. This outcome highlights that touch positions could be readily calculated without relying on resistance acquisition, even under stretched conditions. Notably the method achieves a temporal differential in continuous motions and aligns with the natural routine of hand movements, rather than splitting individual data frames. By utilizing limited information from one-dimensional signals, it decouples bimodal signals while simultaneously accounting for both signal

characteristics and temporal dynamics. Since this approach requires temporal sequence analysis rather than single-time-point signals, it does not support static motion benchmarking. Therefore, this study focused on interpreting hand motions within defined time windows, as it provided the possibility of coupling skin stretch and touch motions in continuous temporal framework.

## Accessible implementation of the HDM metaskin on human skin

To pursue versatile sensory extension, skin electronic interfaces need to adapt to the curvature of the body. The installation of the interfaces should also minimize the hindrance to inherent sensations and somatic movements[37]. Although in-textile photolithography and knitting engineering facilitate the embedded integration of electronic modules in clothes, foreign disturbances and signal loss due to intricate motions remain challenges because the substrates are fully covered[38,39]. The highly conformal adhesion and lightweight configuration of the HDM metaskin provide a feasible scheme. As an action-intensive body region, the index finger was chosen for HDM metaskin attachment for demonstration (Fig. 4a). When interactive movements and stimulation are performed, the integral signal is transmitted via a miniature wireless transmitting module attached to the arm. The high conformability of our metaskin enabled skin-wrinkle-level (10 μm) adhesion on the skin, and contributed to the response to tiny changes. As illustrated in Fig. 4b, the thickness of the substrate covering the skin wrinkles was crucial in the intricate detection of the mechanical strain. The simulation results indicate that a 10 μm substrate effectively captures subtle variations in the wrinkle valleys. In contrast, a 40 μm-thick substrate hindered wrinkle relaxation and tightening, resulting in concentrated strain around the ridge due to increased interfacial shear forces. Moreover, the reduction in film thickness significantly decreases the force required for stretching. This not only minimizes mechanical discomfort but also enhances the seamless and imperceptible implementation of HDM metaskin on the skin (Fig. S7)[40]. By achieving such a lightweight and highly adaptable design, the HDM metaskin serves as an accessible and versatile manipulation device.

The encoded interpretation of finger flexion is commonly determined by two active joints—the proximal interphalangeal (PIP) and metacarpophalangeal (MCP), joints—whose movements can be activated independently by finger muscles and tendons. To minimize motion artifact, sensing units in common strain electronics were installed in isolation on different joints (Fig. 4c). This is because thick films exhibit greater internal tensile forces, which propagate within adjacent regions and caused strain disturbances[41]. In comparison, the HDM metaskin can take advantage of both concise signal and strain adaptation to convey multiple -joint motions precisely. During PIP bending at different angles, the stretching of the conductive path caused apparent resistance changes (Fig. 4d). Even the swing movement of the MCP could be captured, resulting in discernible signal patterns (Fig. 4e). The distinct signal features caused by local joint movements enabled better readability and decoupling for further demonstrations. The exceptional sensitivity of the HDM metaskin allowed for it to decipher the intricate flexion details of all five finger joints from subtle wrist tendon movements (Fig. S14). This proficiency can be further leveraged to interpret converging signals from diverse biomechanical attributes, unlocking new possibilities for advanced proprioceptive monitoring.

From another perspective, finger flexion caused lateral and normal strain, which cannot be distinguished from the pressure-induced strain on the sensor in common layered-structured tactile sensors (Fig. S15). This phenomenon directly caused the crosstalk between touch and strain sensing (Fig. 4f). It caused issues encoding obstacles and pattern similarity, as the normal force introduced pseudo-pressing signals. In contrast, the meta configuration eliminates the possibility of bend-induced compression, enabling direct decoupling of the two types of signals. Owing to the parallel conductive paths being instantly

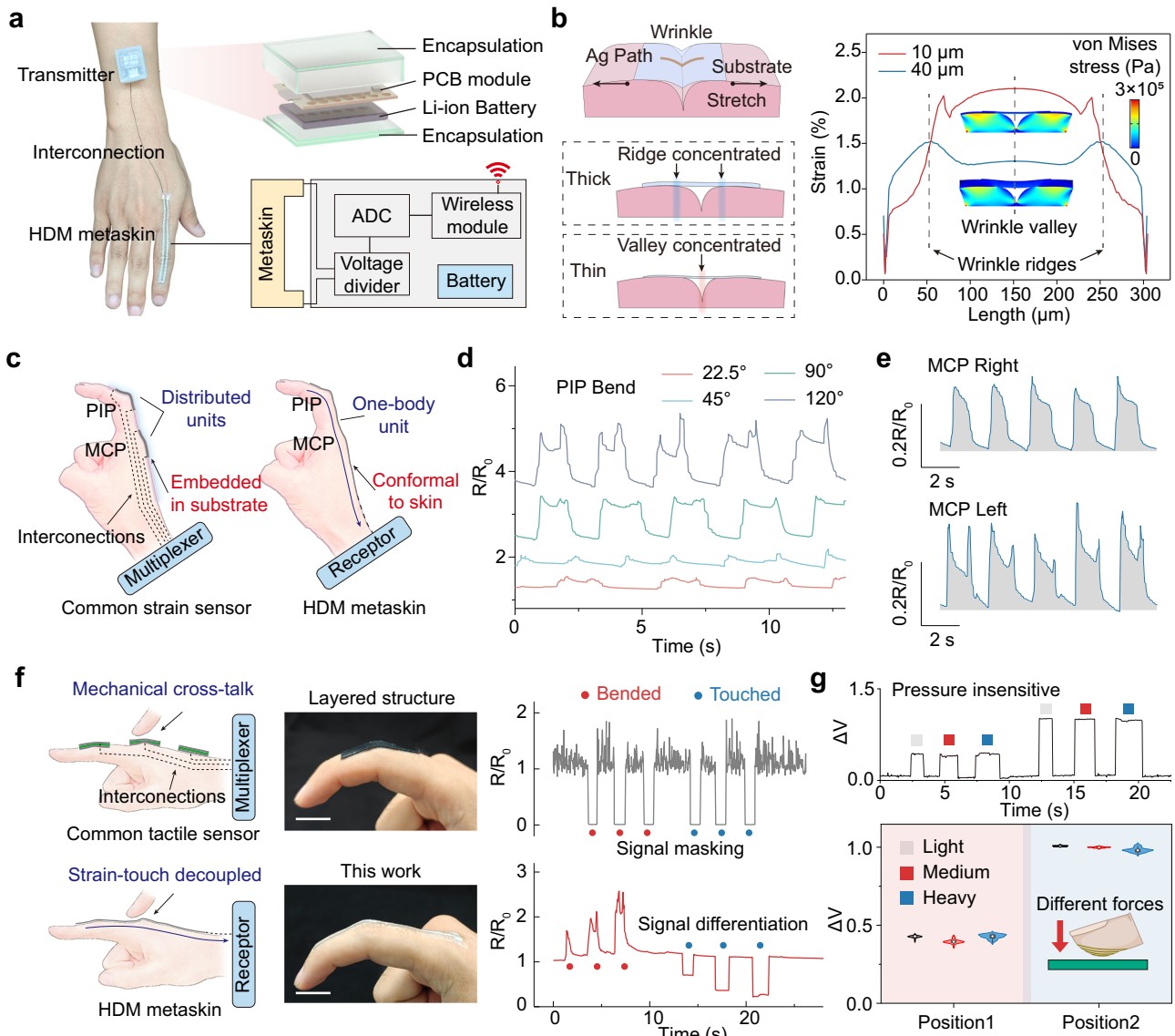

**Fig. 4 | On-skin performance of the HDM metaskin. a** The wireless sensor system, composed of an analog-to-digital converter (ADC), wireless communication module, and battery, all incorporated into a printed circuit board (PCB) with soft Ecoflex encapsulation. **b** Strain of thick (40 μm) and thin (10 μm) film substrate under same stretch force. Finite simulation results show that the thin film can capture and highlight the subtle stretching at the wrinkles, whereas the thick film fails to reflect the strain variation around the wrinkle valleys. **c** Comparison of configurations between common strain sensor and HDM. meta-skin on finger. Common sensors require distributed arrays to monitor the bending of proximal. interphalangeal joint (PIP) and metacarpophalangeal joint (MCP). **d** Signal outputs from PIP joint bends at various angles. **e** Responses of the MCP joint sensor on both left and right movements. **f** Comparison. between common strain sensor and the HDM meta-skin. Detection of joint movements needs different units. To avoid signal crosstalk in common strain sensor design, whereas thin-film substrate and in-plane. configuration avoid stress interference between regions. Scale bar: 2 cm. **g** Influence of pressing intensity. on the positioning signals at two different touch positions, showing insensitivity to the light, medium, and heavy force.

saturated, the voltage changes only responded to the touch position during touch using different forces (Fig. 4g). Moreover, as a conceptual extension, the pressure-sensitive implementation was also verified using a striped-pattern (Fig. S16).

## Signal decoupling and encoding for contextual tasks

Somatic sensation coordination is indispensable for exploring one's surroundings and executing bodily actions. Despite the numerous publications on functional electronics, research on effectively interpreting body motions remains scarce, especially those involving multimodality devices[42] (Fig. S17). In a typical scenario, a person needs to adjust their body posture and hand position to align with context demands for effective object handling (Fig. 5a). This often requires asynchronous acquisition with multichannel sensors to integrate

motion with bodily contexts, which can lead to spatiotemporal misalignment. The data-fusing pattern of the HDM metaskin effectively expands the information space, enabling the reconstruction of bodily motion awareness through temporal differentiation (Fig. S18). Here, a system bridging intrinsic body awareness and extrinsic contextual interaction was constructed, enabling instinctive and adaptive control of robotic behaviors. Owing to its customized printing ability, the HDM metaskin achieved programmable shapes to conform to different body regions and tasks. For the wrist, we choose a U-shaped configuration to both increase the number of interaction points and enhance the visual demonstration effectiveness. When attached to the wrist, the U-shaped HDM metaskin simultaneously monitored the hand posture and touch stimuli, serving as the context and motor command inputs, respectively. These merging signals were then processed by filtering

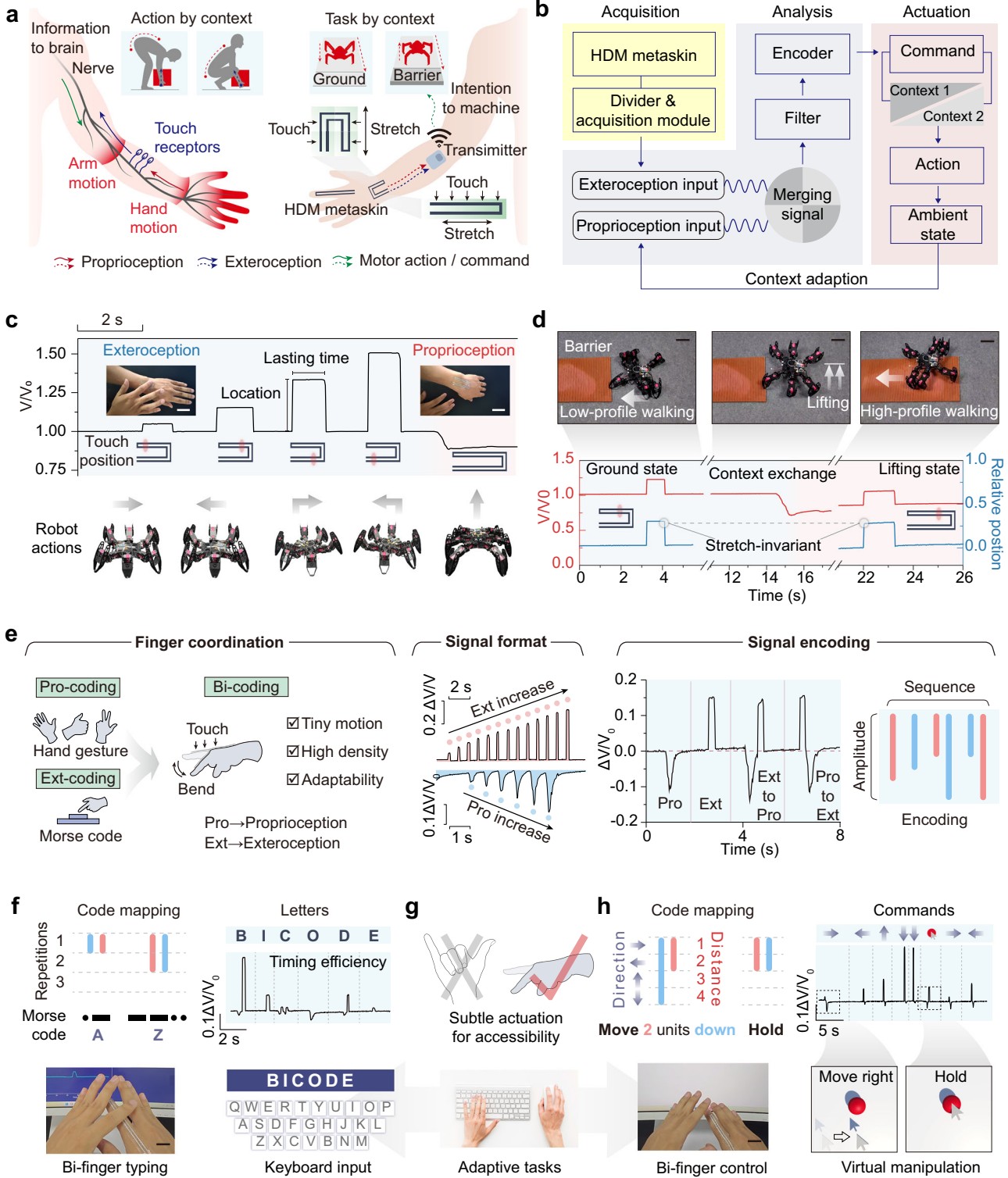

**Fig. 5 | Navigation and manipulation applications informed by proprioceptive contexts. a** Depiction of the human somatosensory system and its artificial analog. The intact perceptual loop enables humans to process tasks in various contexts, and adjust to varying operational environments. **b** Overview of the system architecture, detailing modules for data acquisition, computation, and command generation based on contextual inputs from proprioception and exteroception. **c** Mapping of proprioceptive and exteroceptive movements to five different robot commands using a U-shaped meta-skin installed on the wrist. Scale bar: 5 cm. **d** Adaptation of a hexapod robot to different ground and lifting profiles, navigating both barrier and nonbarrier contexts on the floor. Scale bar: 10 cm. **e** Bi-coding method integrating proprioceptive and. exteroceptive finger motions. **f–h** Application of the bi-coding strategy in typing and mouse control. The bicoding strategy enables rich information encoding through small motions, supporting complex interactive. tasks for disabled users. Scale bar: 2 cm. Contains icons via Freepik License.

and encoding, providing dynamic and contextual cues for task refinement (Fig. 5b). For example, a motor robot requires to adjust its moving posture and moving speed under variable terrain conditions. As shown in Fig. 5c, the system interpreted the touch positions to navigate the walking directions of a hexapod robot, and the wrist bending states mapped to the walking postures—wrist flexion causing lifting of the machine body. The robust decoupling of bimodal signals allowed for us to relocate the command index position even in different wrist bending states. By harnessing the information abundance in a single device, we elaborated the adaptive interaction in a live navigation task (Fig. 5d and Supplementary Movie 2). The walking profile of the robot is correlated with the hand bending state, and under different command contexts, the robot moves with an appropriate stride to align with the real environment. In another example, precise navigation of the hexapod robot was demonstrated using a linear meta-skin, highlighting the expanded capabilities in both form and function of meta-skins (Fig. S19 and Supplementary Movie 3). As shown previously, the decoupling calculation of the touch position was not affected by the stretching state, allowing for accurate positioning of the relative touch position even when the context changed. Therefore, the HDM metaskin exhibited an extended ability to reactively and anticipatorily recognize the interactive intentions of humans, as naturally as a skin extension.

Interpretation of body motions enables the expression of behavioral information such as body signs and hand gesture language[43,44]. Hand motion encoding was used for effective intention representation in the case of speech disorders, requiring a balance between high informational density, ease of interpretation, and semantic alignment[45]. The bimodal signal of the HDM metaskin enabled easy triggering through subtle finger flicking when it was installed on the finger. This eliminates the need for an extensive motion corpus, as is required in conventional hand gesture encoding, and addresses the limited information capacity and adaptability of methods such as Morse code[46] (Fig. 5e). Specifically, owing to the continuity of the finger flick and the touch positions, the signal format exhibited both binary characteristics and continuous differentiability. By combining the signal sequence and amplitude, the finger motions can be encoded into two-dimensional digital information. We achieved adaptive tasks by equipping the finger encoding system, to simultaneously simulate virtual typing and mouse control (Fig. 5f-h). Notably, the bi-coding scheme requires the coordination of only two index fingers with minimal motion range, making it suitable for individuals with developmental motor disorders (Fig. 5g). The typing encoding was a marked improvement from Morse code, which relies on time allocation to encode additional informational dimensions. In the single-channel signal sequence, bi-coding uses bidirectional signaling to replace time-incorporating binary encoding. The signal amplitude was divided into three segments to express the input number of the binary index. This means that a light finger flick corresponds to one symbol of "−", a medium flexion corresponds to "− −", whereas a large finger flexion output corresponds to "− − −". For example, a light flick along with touching the bottom of the finger collectively represented the sequence "-...", which consequently indicated the letter B. The bi-coding approach combines high information density with optimal timing efficiency. Herein, we showcased the signal format and Morse code interpretation of the word "BICODE" (Supplementary Movie 4) to highlight its capabilities and timing efficiency. The complete mapping list of all 26 English letters is provided in Fig. S20. Additionally, our method was able to encode the mouse operation through logically defining movement and holding commands (Fig. 5h). Specific commands are executed as motifs, which are determined by the combination of motion patterns (proprioception or exteroception), and signal amplitude levels, which correspond to the movement direction and distance. In the live demonstration, we successfully manipulated the pointer to grab and move a virtual ball (Supplementary Movie 5).

The above applications demonstrated that our encoding mechanism can effectively integrate bidirectional signaling with minimal motion requirements, enabling multiple tasks and seamless switches. This inspired a low-cost and imperceptible skin interface designed for individuals requiring precise yet low-effort interaction methods.

## Contextual hand–object interaction tasks with machine learning

Machine learning helps to expand the application of electronic devices to various scenarios beyond simple motion encoding, including more complex tasks such as object recognition and interaction[47,48]. However, this advancement faces significant challenges in terms of robustness, especially when dealing with ambiguous and similar motion classification, making it more difficult to achieve consistent performance across diverse conditions. Moreover, owing to the continuous generation of motion frames, proprioceptive sensors struggle to filter out unintended motions and artifact effectively[49]. In the context of gesture classification, unintended or unconscious hand movements can often trigger predefined labels, despite not reflecting the user's intent. This poses a significant challenge in generalization, where the model struggled to differentiate unintentional movements and body contexts. To address the above issues, we demonstrated a general hand–object interaction framework based on contact context and proprioception movement. The perception network consisted of the HDM metaskin attached to the palmar side and dorsal side of the finger, and the wrist (Fig. 6a). They were used to monitor object-holding states and hand motions, with data collected through separate signal channels for each. Since object interaction events are decomposed into combinations of contact states (such as no contact with the object and different postures of contact) and joint actions (e.g., action direction and amplitude), most object interaction events can be accurately described through the combination of proprioceptive signals and contact signals. Notably, although contact states, as a necessary condition for interaction context judgment, are difficult to distinguish through machine learning, they provide a stable boundary that helps define the domain of machine learning classification. Thus, the proposed framework leveraged the ability to substantially differentiate hand posture through contact signals, creating a reactive system characterized by event boundaries. In other words, the exteroceptive touch state offered the precondition of body context, and the proprioceptive signal was applied for classification prediction assisted with deep learning. Within distinct feature contexts, the requirement for system generalization was alleviated, rendering it particularly well-suited for systems operating with few datasets.

In the learning phase, a convolutional neural network (CNN) assisted long short-term memory (LSTM) network was adopted to train and classify the hand motions (Fig. 6b). To increase the recognition accuracy, we introduced mathematical operators (derivative, difference, Hilbert, and entropy operations) for feature augmentation prior to training (the detailed calculation process is analysed in Note S3). To do this, we divided the time sequence data into fixed-length sliding windows as inputs and tested the above four methods to generate additional aligned sequences (Fig. S21 shows the model performance for various time-window sizes). A CNN layer was used in the feature extraction phase to capture local correlations, offering structured information that compensates for the limitations of LSTM in modeling short-term dependencies. For time sequence learning, the data were trained in multiple LSTM layers to analyse long-term dependency features, aiming to achieve better generalization in learning tasks. The dense layer subsequently transforms the input features into a higher-dimensional space to learn complex mappings between inputs and outputs, whereas the dropout layer is applied as a regularization technique to reduce overfitting.

The HDM metaskin allowed for participants to achieve contour recognition through continuous probing and rubbing, such as identifying geometric shapes on the basis of variations in hand flexion. Fig 6c

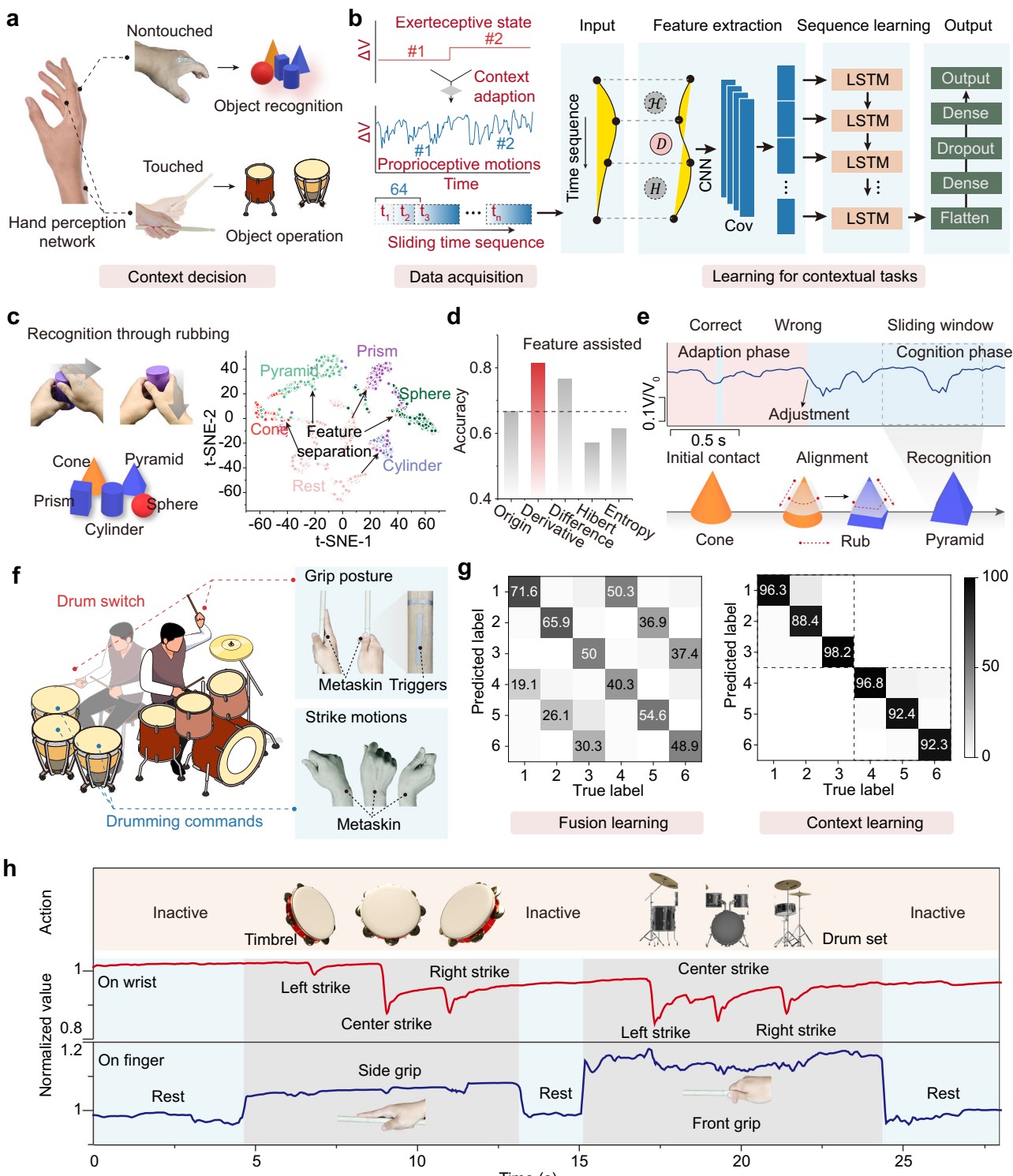

**Fig. 6 | Object interactions informed by touch contexts. a** A contextual object interaction framework determined by touch state and touch position. Contains photos by New Africa via Freepik License. **b** Machine learning assisted data acquisition, feature extraction, and sequence learning to integrate and classify proprioceptive and exteroceptive signals. *CNN* convolutional neural network; *LSTM* long short-term memory; *Cov* convolutional layer) (**c**) Method for object recognition using proprioceptive feel, and the t- SNE mapping of recognition signals in a feature space. **d** Classification enhancement with feature extraction. **e** Dynamic recognition process involving phases of initial contact, proprioceptive alignment, and comprehension during object recognition. **f** Context-switching drum performance application, where drum classes are switched by changing the holding posture of the drumstick, and playing is controlled by recognizing different wrist motion patterns. Contains icons by macrovector via Freepik License. **g** Confusion matrices comparing fusion learning and context boundary method for similar wrist motion classification. **h** Signal analysis of the playing process, where drum switching and playing are controlled by separate signal channels. Contains photos by isometricworld and rafael-19santos via Freepik License.

shows the geometric objects with distinct shapes selected for the experiment, along with the process of recognizing these shapes through palm exploration. When the person rubbed the object with their hand, proprioceptive flexion signals encoded contour-related features. In the feature space, the characteristics of the five geometric shapes, represented by different colors, diverged from those of the resting state in distinct directions, projecting into dedicated regions. Compared with direct training, differential and gradient-based computations involving local variation rates significantly improved classification outcomes, with derivative feature enhancement increasing the recognition rate from 66% to 81% (Fig. 6d). However, enhancement methods involving Hilbert transformations and entropy calculations, which involve frequency-related information, adversely affected the classification results (Fig. S22). Figure 6e shows the exploratory recognition process for a pyramid. Like how humans explore the environment, object recognition does not establish confidence at the initial contact stage but requires an adjustment period, thereby limiting the upper bound of cognitive accuracy to some extent. During the adaptation phase, the hand's adherence and rubbing movements lack stability, leading to confusion between the pyramid and the cone. After a second adjustment and regrasping, the final recognition result was corrected to the pyramid. In another example, the HDM metaskin demonstrated its ability to decipher intricate joint flexion and directional cues within single-finger movements through time-series learning, as evidenced by typing on a numpad (Fig. S23). Specifically, vertical click actions were decoded as different combinational patterns of PIP and MCP joint flexion, discernible from the spike composition in the sequential signal spectrum. Moreover, lateral click actions corresponded to the lateral sway of the MCP joint, which modulated the peak amplitudes. These motion patterns are reflected in the signals and ultimately enable precise and reliable motion classification.

We aimed to provide a natural interaction mode that adapts to bodily movements through sensing modality fusion. To this end, a demonstration of air-drumming was shown, which featured a switchable definition boundary for limited training labels, making it possible to achieve smooth contextual interaction on the basis of biomimetic action habits. As shown in Fig. 6f, wrist motions in three directions simulate the air drumming actions, captured by the wrist-mounted sensors, whereas the finger-mounted sensors monitor the drumstick grip posture, controlling the switch of the drum type. The drumstick featured a T-shaped silver composite conductive pattern, serving as a trigger for the HDM metaskin to respond to contact positions. The first grip posture, called the open-grip posture, generates the sound of a snare drum in the virtual application, whereas the second posture, the closed-grip posture, corresponds to the sound of a hand drum. As illustrated in Fig. 6 g, after training on wrist movements (left, center, right) for both grip postures, the model exhibited significant confusion in classifying six distinct actions (labels #1 to #6). This confusion arose because wrist movement signals do not noticeably change with different grip postures, making the accurate extraction of static features using machine learning methods nearly impossible. Thus, to coordinate the matching of the interaction posture and dynamics, multi-dimensional information from both the grip posture and wrist movements must be considered simultaneously. We addressed this issue in a parallel processing framework, where contact position signals were used as conditional triggers to limit the activation space of the learning model (Fig. S24). As a result, even with training limited to wrist movements under the open-grip posture, the model effectively operated for the closed-grip posture, achieving an output accuracy exceeding 90%. This approach significantly reduced the definition and learning costs. Details of the demonstration process are shown in Fig. 6h and Supplementary Movie 6. Initially, no output was produced when the hand did not make contact with the trigger, conserving computational resources by filtering out invalid action frames. In the open-grip posture, the predictive model was activated and responded solely to three hand drum trigger sound labels. In the closed-grip posture, the model successfully predicted three snare drum trigger sounds. Unlike the object shape prediction outputs, the sliding window count in our model did not overlap, meaning that the output was not continuous but responded only to individual wrist-triggered actions. This application demonstrated that our HDM metaskin system could concurrently interpret both interaction postures and dynamics between the hand and object, aligning with contextual object interactions under natural bodily movements.

## Discussion

Based on the understanding of bodily dynamics, we present an encapsulation-free artificial mechanoreceptor, termed an HDM metaskin, which conforms closely to the skin topography and accurately decodes proprioceptive motion and exteroceptive contact. The two motion modalities exhibit dual-polarity signal patterns that are individually salient, thereby eliminating mechanical crosstalk. Our method establishes temporal differential decoupling, compares the temporal differences in sequential signals, and deciphers the contextual correlations between proprioceptive-tactile dual-mode signals. This method enables the precise positioning of relative touch positions even under stretching conditions. The strong interfacial coupling between the thin hydrophilic substrate and the water-based Ag nanocomposite is the physical basis of the mechanoreceptor, with a two-step transfer strategy to minimize interfacial losses and uneven strain. The results revealed that the HDM metaskin exhibitedhigh conformability and stability, because the scale between the HDM metaskin and the skin wrinkles (~10 μm) was identical. Under experimental conditions, the intrinsic resistance of the nanocomposite, which was placed on the fingertip, remained nearly unchanged during continuous friction (2 N for 10 min), during exposure to high humidity (90% for 2 h), and within a human-comfortable temperature range (25–60 °C). In addition to responding to skin stretches, the HDM metaskin provides a perspective on touch positioning using imperceptible conductive triggers, thus taking contact dynamics into consideration. Proprioceptive and exteroceptive components in the time-series signals convey motion details and behavioral intents through superposition, sequence, and parallel combinations. The time patterns of dynamic motions can be compiled into time-efficient logical information, simulating keyboard and mouse operations to aid individuals with motion impairments. This implies an application adaptation to bodily contexts, overcoming the limitations of conventional sensor systems, which often lack flexibility and are restricted to predefined programming. Additionally, demonstrations revealed that the directionality and angular cues of finger joint movements could be extracted by deep neural networks, expanding the physical boundaries for mapping explicit intent in hand movements. Furthermore, the feasibility of soft tissue surface pressure sensing on the basis of the HDM metaskin was also experimentally confirmed, which compensates for simplifications of pressing actions, and guides future research. Future projects lie in deploying the metaskin across all five fingers and other joints to acquire comprehensive motion data, driving applications in soft robotics, embodied interactions, and intelligent prosthetics.

## Methods

### Preparation of the water-based Ag NP/Ag NW nanocomposite paste

First, we dispersed 10 mL of an Ag NW solution (10 mg/mL; diameter: 90 nm; length: 100 μm; Aladdin) using ultrasonic sonication (550 W) for 10 min. The mixture was then centrifuged at $850 \times g$ (3000 rpm) for 10 min. To obtain a concentrated Ag NW solution, we carefully removed 9 mL of the supernatant with a pipette. The concentrated solution was sonicated again for 10 min. Two grams of premade Ag nanoparticle paste (50%Ag NPs: 50%PU, Mogan Technology) was subsequently added to the Ag NW solution and stirred until thoroughly

mixed. The final proportion of Ag NPs to Ag NWs was 10:1. As a control, multilayered carbon nanotubes were also mixed with Ag nanoparticle paste in the same proportion to prepare the Ag NP/CNT paste (CNT, 2 mg/mL; Times nano). For all the nanocomposites involved in the experiment, the major solvent was deionized water.

## Preparation of the HDM metaskin

HDM metaskin was achieved through the fabrication of thin films and the printing of sensitive Ag paths, followed by a secondary transfer process. Initially, a high-concentration waterborne polyurethane (WPU) solution was spin-coated onto transfer paper at a speed of 3000–4000 rpm, producing a film with a thickness of ~4–10 μm. The coated film was then dried on a hot plate at 40 °C for 10 min. Subsequently, an Ag nanocomposite paste was precisely deposited onto the film surface using a dispensing and printing system (DB100, Shanghai Mifang Electronic Technology Co. Ltd.) at a print pressure of 60 MPa and a speed of 10 mm/s. The printed film was oven-dried at 60 °C for 2 h.

In the transfer stage, the WPU film was cut to the desired dimensions and laminated onto a silicone rubber substrate. The mild tackiness of the WPU film facilitates its adherence to the silicone, which has a slightly lower surface energy, enabling easy detachment from the transfer paper. A thin layer of pressure-sensitive adhesive (PSA) was then spin-coated onto the exposed WPU surface and subsequently dried on a hot plate at 50 °C for 10 min. The adhesive-coated side of the film was pressed onto the skin, and the silicone substrate was peeled off, enabling a strain-free transfer of the HDM metaskin onto the skin. Finally, the conductive Ag nanocomposite pathways were exposed on the surface.

## Characterization and electrical measurement

Scanning electronic microscope (SEM) characterization was conducted with a field emission electron microscope (SUPRA 55 SAPPHIRE, Carl Zeiss). Tensile tests of films were performed with a universal testing machine (ZQ-990B, Dongguan Zhiqu Co., Ltd.) at a measurement rate of 40 mm min$^{-1}$. Resistance signals were measured on an LCR meter (TH2840, Changzhou Tonghui Electronic Co. Ltd.) at an AC voltage of 1 V and a sweeping frequency of 1 kHz. Voltage conversion signals were measured and recorded using a digital multimeter (34465a, Keysight). The thickness of the thin films was measured using an automatic stylus profiler and stress measurement system (Dektak XT-A, Bruker) with a vertical scan range of 65.5 μm. Power consumption measurement was conducted with a direct current power analysis instrument (PowerScope, Nanjing Sensingsystmes Co. Ltd.).

## Mechanical simulation of a film on skin

The strain and stress distributions of the thick and thin-film electronics were compared through the finite element method (COMSOL Multiphysic, version 5.6). The width and depth of the skin wrinkles were set as 160 μm and 80 μm, respectively. A hyperelastic WPU material model was applied above the wrinkle, and the thicknesses of the thick film and thin film were set as 40 μm and 10 μm, respectively. The Ag nanocomposite conductive path was set at a thickness 10 μm. The strain curve was obtained by extracting the strain data from the top surface of the Ag conductive path.

## Wireless communication module

The wireless measuring module was developed on the Xiao ESP32C3 (Seeed Studio) development board to transmit Wi-Fi signals and relay data. The whole system consists of an analogue-to-digital converter (ADC) sensing element, a volage divider element, a Wi-Fi module, and a lithium-ion battery. The device was configured in station (STA) mode, in which it gathers analogue signals from connected sensors and transmitting the data to the computer over Wi-Fi. The analog signals are converted into digital forms through the analogue-to-digital converter interface, and the collected data is sent every 30 ms. Python (version 3.13.0) was used to handle incoming data with multithreaded programming, ensuring real-time updates and enabling further data processing (Supplementary Movie 7). The wireless communication module operates in three power states with the following average power consumption: the unassociated state (0.079 W), the associated idle state (0.160 W), and the active data transmission state (0.302 W) (Fig. S25).

## Implementation and control of the hexapod robot

A U-shaped HDM metaskin was affixed to the user's wrist, with a wireless module connected to its terminal. Single-channel data comprising somatic action information were transmitted to a personal computer (PC) for further differentiation processing. A software interface developed in Python collected these data and managed the conversion and transmission of control signals. The communication system utilized a pair of HC-05 Bluetooth modules configured in master–slave mode to establish a reliable wireless connection between the PC and the hexapod robot. The robot's locomotion was concurrently governed by both the contextual and addressing features of the received signals. With the use of a decoupling algorithm, the relative touch location on the HDM metaskin remained consistent despite wrist flexion. On this basis, identical touch positions resulted in varied moving parameters (e.g., moving distance as demonstrated in Supplementary Movie 2) depending on the t bodily context (wrist flexion state).

## Neural network learning

For different tasks, interaction actions were continuously collected in a sequential acquisition mode (ranging from 30 to 100 repetitions) to obtain training and testing datasets. Specific gesture actions were labeled with the same action tag to serve as supervisory signals for classification. We developed a deep learning framework tailored for time series classification. Feature extraction was performed by calculating the derivatives of the sensor signals to enrich the dynamic characteristics. Additionally, to enhance feature selection, we employed differential operations, Hilbert transforms, and entropy calculations to supplement features with diverse attributes. All the data were standardized using z-score normalization to ensure uniform scaling. A sliding window approach was used to segment the normalized data into continuous, fixed-size sequences. The time series classification model integrates a one-dimensional convolutional layer (Conv1D) to extract local temporal patterns, followed by two long short-term memory (LSTM) layers to capture temporal dependencies. The network architecture included fully connected dense layers and dropout layers to mitigate overfitting. In practical interaction classification tests, two methods—continuous prediction mode and frame judgment mode—were adopted to accommodate different classification output requirements. For example, in object recognition tasks that involve continuously adjusting hand poses to achieve comprehensive judgment, prediction results for long-duration action windows were output consecutively. Conversely, for drum action prediction tasks that require a one-to-one correspondence between single actions and single labels, only individual windows within single action segments were collected for classification. Importantly, this approach relies on high prediction accuracy in model training, necessitating adjustments in dataset quality and model training parameters to achieve optimal performance.

## Statistical analysis

Error bar analysis in this work represent mean ± standard deviation (SD), showing central tendency of the data. Box plots analysis in this work were used to represent the distribution of the data. The central line in each box represented the median, the lower and upper edges of the box indicated the 25th and 75th percentiles respectively

(interquartile range, IQR), and the whiskers extended to the minimum and maximum values within 1.5×IQR from the quartiles. Unless otherwise specified, all statistical samples in this experiment were derived from independent material samples or tests. The experimental data were analyzed with Matlab (version R2022b) and Origin (version 2017).

## Ethics oversight
All procedures involving human research participants were conducted in accordance with the experimental protocol approved by the Ethics Committee of Xiamen University (XDYX202501K02). All participants were informed with written consent.

## Reporting summary
Further information on research design is available in the Nature Portfolio Reporting Summary linked to this article.

## Data availability
The data support the findings of this study is available within in the main text and the Supplementary Information/Source Data file. The raw training data for machine learning are included in the provided code. Source data are provided with this paper.

## Code availability
The python code (Python 3.9) for the CNN-LSTM model and relative algorithm are publicly available on Github/Zenodo[50] at (https://doi.org/10.5281/zenodo.17060795) along with the paper.

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

## Acknowledgements

This work was supported by National Key Research and Development Program of China (2024YFA1208204 and 2024YFB3816200 to X.L.), National Natural Science Foundation of China (52202117 to X.L.) (52232006 and 52072029 to Q.L.), Fujian Provincial Natural Science Foundation of China (2025J010012 to X.L.), State Key Laboratory for Advanced Metals and Materials (2025-Z02 to X.L.), 2023 Key Project of the Ministry of Industry and Information Technology (246 to Z.C.), and the Ministry of Education, Singapore, under its MOE ARF Tier 2 (MOE-T2EP30123-0019 to Y.Z.). Any opinions, findings, conclusions, or recommendations expressed in this material are those of the author(s) and do not reflect the views of the Ministry of Education, Singapore.

## Author contributions

Conceptualization: S.Y., X.L., and Z.C.; Supervisor and fund support: Z.C., X.L., Y.Z., and Q.L.; Experiments: S.Y., Z.J., L.L., Z.H., Y.L., H.W., and R.W.; Discussion: S.Y., Z.J., Z.G., and X.L.; Software: Z.J., Y.L., and H.W.; Data analysis: S.Y. and Z.H.; Original draft: S.Y. and L.L. All the authors participate in the review and editing.

## Competing interests

The authors declare that they have no competing interests.
