## [Transparent Peer Review file · Nature Communications]

A hyperconformal dual-modal metaskin for well-defined and high-precision contextual interactions

Corresponding Author: Professor Xinqin Liao

Version 0:

Reviewer comments:

Reviewer #1

(Remarks to the Author)

The article, "A hyperconformal dual-modal metaskin for well-defined and high-precision contextual interactions", presents an interesting and enlightening concept: decoupled "bodily context" to expand hand movement interpretation. Inspired by human's proprioception and touch senses, the authors successfully combined hand stretching and touch positioning sensing modes into a hyper-conformal electronic skin with subsequent demonstrations to prove the concept. Specifically, they initiated a two-step transfer methodology, which is effective to solve the problems of conformal strain sensing and touch trigger. This light-weight design of the metaskin allows the unobstructed interaction and simplifies the interaction burdens. Finally, the detailed applications, including robot control, keyboard and mouse simulation, object recognition, and drumming performance, verify the significant potential of this study in future wearable electronics.

Overall, the manuscript is well organized. The mechanism and fundamental theory were clearly explained. The motivation is clearly demonstrated. Also, the data graphs and illustrations are exceptionally well done. The videos are well filmed to show the details of preparation process that is important for follow-up research in a reproducible behavior. Based on above, I recommend acceptance if they could address the concerns below, which could help refine the manuscript even further.

1. I note that the proposed temporal differential decoupling method is derived through cross-temporal comparison rather than real-time computation. The authors should discuss the potential influence of this approach in the manuscript.
2. The performance visualization in Fig. 2b lacks readability. The X-axis could be replaced with the damage severity rather than the substrate type for clearer interpretation.
3. Given the considerations of portability and energy management, the authors should provide detailed test data of the system power consumption.
4. In Fig. 3g, the output voltage is controlled within 5 V. The authors should provide necessary explanations.
5. In Fig. S12b, the word "Oberseved" may need further check.

Reviewer #2

(Remarks to the Author)

This paper presents a development of hyperconformal dual-modal metaskin for interactive hand motion interpretation by integrating a strongly coupled hydrophilic interface with a two-step transfer strategy to minimize interfacial mechanical losses.

This study is expected to provide a useful wearable sensor platform and this may be recommended for publication if the authors can successfully respond to the following comments,

- 1) The authors said to 'integrate both perceptual modes within the temporal framework, two key challenges must be addressed: achieving high-precision differentiation between them and capturing their coupling relationship in continuous motion. To better describe this relationship, a concept named "bodily context" is introduced, focusing on context estimation of motions besides the discrete definition. The concept of 'bodily context' is still not easy to understand and it had better be further explained.
- 2) Due to the strong adhesion between the nano-composite paste and the WPU substrate, the HDM metaskin transferred to the skin exhibits exceptional electrical and mechanical robustness. It maintains stable electrical resistance under prolonged frictional wear, remains unaffected by high humidity (90% RH), and retains thermal stability across a wide temperature

range. These attributes ensure reliable performance in diverse and dynamic environments (Fig. 2e). Usually, metal shows a PTC behavior where the resistance increases as the temperature increase. I wonder why the metaskin does not show temperature dependence.

3) Figure 5 shows mapping of proprioceptive and exteroceptive movements to five different robot commands using a U-shaped meta-skin installed on the wrist. The system interpreted the touch positions to navigate the walking directions of the hexapod robot, and the wrist bending states mapped to the walking postures—wrist flexion causing lifting of the machine body. The robust decoupling of bi-modal signals allowed us to relocate the command index position even in different wrist bending states. The walking profile of robot correlated to hand bending states, and under different command contexts, the robot would move with an appropriate stride to align with the real environment. It seems to me that the robot can do a specific job depending on the location of touch. I wonder if the command can be more sophisticated (for example, the move distance, angle, speed etc).

4) The metaskin leverages the integration of the ultrathin hydrophilic substrate (8–10 μm) and highly compatible water-based Ag nanocomposite sensitive path, minimizing impediments on hand movements. However, 8-10 μm film had better be called thin substrate. Ultrathin e-skin usually refers to the sub-micron thin film substrate.

5) From the video clips for the hexapod robot, typing, mouse control and drumming, the response seems to be a bit delayed. What is the limiting factor for delay and how can it be improved?

6) Various soft e-skin sensors for hand motion recognition have been developed recently. Very recently, nanomaterial based soft sensors with machine learning for hand motion detection (Nat. Electron., 6, 64, (2023); Nat. Comm., 11, 2149, (2020)) have been demonstrated for virtual keyboard/touch and shape recognition. They should be briefly introduced in the manuscript and the novelty compared with those recent studies needs to be discussed.

7) Hand gesture plays a very crucial role in human-machine interface and VR interface. Hand gesture recognition for interfacing with VR application (Adv. Funct. Mater., 31, 2007772 (2021); Adv. Funct. Mater., 32, 2106329, (2022); Natl. Sci. Rev., 11, nwad298 (2024)) needs to be briefly discussed in the introduction part to help the readers with the recent related progresses.

8) A U-shaped HDM metaskin was affixed to the user's wrist, with a wireless module connected to its terminal. Single-channel data comprising somatic action information was transmitted to a personal computer for further differentiation processing. A software interface developed in Python collected these data and managed the conversion and transmission of control signals. The communication system utilized a pair of HC-05 Bluetooth modules configured in master-slave mode to establish a reliable wireless connection between the PC and the hexapod robot. I wonder why U-shaped configuration was used for this application. Does it require long length?

9) The general usage of language including typos and grammatical errors need to be checked again.

10) Some of the digital pictures are missing scale bars. They should be added to the pictures.

Version 1:

Reviewer comments:

Reviewer #1

(Remarks to the Author)

In this version, the authors have provided additional experiments to address the concern, including experimental results and the supporting of characterization system. The concerns have been clarified, and the discussion is well organized to support the claims in the manuscript.

I would thus recommend acceptance of the current work in Nature Communications.

Reviewer #2

(Remarks to the Author)

The authors responded well to the comments.

Comments from Reviewer #1:

The article, “A hyperconformal dual-modal metaskin for well-defined and high-precision contextual interactions”, presents an interesting and enlightening concept: decoupled “bodily context” to expand hand movement interpretation. Inspired by human's proprioception and touch senses, the authors successfully combined hand stretching and touch positioning sensing modes into a hyper-conformal electronic skin with subsequent demonstrations to prove the concept. Specifically, they initiated a two-step transfer methodology, which is effective to solve the problems of conformal strain sensing and touch trigger. This light-weight design of the metaskin allows the unobstructed interaction and simplifies the interaction burdens. Finally, the detailed applications, including robot control, keyboard and mouse simulation, object recognition, and drumming performance, verify the significant potential of this study in future wearable electronics.

Overall, the manuscript is well organized. The mechanism and fundamental theory were clearly explained. The motivation is clearly demonstrated. Also, the data graphs and illustrations are exceptionally well done. The videos are well filmed to show the details of preparation process that is important for follow-up research in a reproducible behavior. Based on above, I recommend acceptance if they could address the concerns below, which could help refine the manuscript even further.

Response: We are very grateful to you for the affirmation and recommendation of the work. All your comments and suggestions are very helpful to improve the presentation of our findings. The paper is carefully revised according to your great comments and suggestions as follows.

1. I note that the proposed temporal differential decoupling method is derived through cross-temporal comparison rather than real-time computation. The authors should discuss the potential influence of this approach in the manuscript.

Response: We sincerely appreciate your insightful comments, which significantly improve our manuscript. The temporal differential decoupling method calculates action changes based on preceding signal states, and is developed based on the first principles of hand movement biomechanics. This approach decouples bimodal signals by leveraging limited one-dimensional signal information while simultaneously characterizing both signal properties and temporal dynamics. As the method requires temporal sequence analysis rather than single-time-point signals, it does not support static motion benchmarking. Given it focused on dynamic behavior characterization, our

study specifically discusses hand motion interpretation within time windows.

According to your helpful comment, we add more descriptions on Page 9 (last paragraph) of the revised manuscript in red font.

Amended in:

Page 9 (last paragraph): “By utilizing limited information from one-dimensional signals, it decouples bimodal signals while simultaneously accounting for both signal characteristics and temporal dynamics. Since this approach requires temporal sequence analysis rather than single-time-point signals, it does not support static motion benchmarking. Therefore, the study focused on interpreting hand motions within defined time windows, as it provided the possibility of coupling skin stretch and touch motions in continuous temporal framework.”

2. The performance visualization in Fig. 2b lacks readability. The X-axis could be replaced with the damage severity rather than the substrate type for clearer interpretation.

Response: Thank you so much for your insightful comment. According to your kind suggestion, we revise the graph type of Fig. 2b to increase the readability. The relevant statement is also revised on Page 6 (second paragraph) of the revised manuscript in red font.

Amended in:

Page 6 (second paragraph): “The water-based silver paste exhibited inadequate abrasion resistance on hydrophobic substrates, as evidenced by substantial degradation on Ecoflex and SEBS following only a few peeling cycles. In contrast, the silver patterns deposited on the WPU substrate retained their structural integrity with only negligible damage, even after enduring 200 consecutive peeling cycles.”

Revised Figure:

Fig. 2b. Adhesion durability of printed Ag paths on different substrates under cyclic peeling tests. The Ag paths on WPU withstand the highest number of peeling cycles while showing minimal damage.

3. Given the considerations of portability and energy management, the authors should provide detailed test data of the system power consumption.

Response: We are very grateful to you for taking time to provide such thoughtful feedback. According to your great comment, we supplement a consumption test in Fig. S25 (Supplementary Information). We also add relevant statement and method introduction on Page 18 (second paragraph) of the revised manuscript in red font.

Amended in:

Page 18 (second paragraph): “The wireless communication module operates in three power states with the following average power consumption: the unassociated state (0.079 W), the associated idle state (0.160 W), and the active data transmission state (0.302 W) (Fig. S25).”

Added Figure:

Fig. S25. Power consumption of the portable sensor system.

Working power consumption at three different connection states are recorded, including (a) unassociated state, (b) associated idle state, and (c) active data transmission. In the unassociated state, the power consumption (0.079 W) is primarily contributed by the microcontroller unit (MCU), analog-to-digital converter (ADC) modules, and the HDM sensor. When associated, the power consumption increased to 0.160 W and 0.302 W, respectively.

4. In Fig. 3g, the output voltage is controlled within 5 V. The authors should provide necessary explanations.

Response: Thank you very much for your thoughtful comment. Indeed, the stretch and touch position induce opposite resistance variations. Their intensity cannot be decoupled due to shifting resistance baselines. To quantify the relative variation, the output voltage is constrained within a fixed interval. The converted voltage signal will only change predictably within this range. The stretch can be quantified by compared the voltage before ($V_{s,0}$) and after ($V_{s,t}$) stretching:

$$S_t = \frac{V_{s,t}}{V_{s,0}}$$

Through proportional scaling, the relationship between touch position (P_t) and contact voltage ($V_{c,t}$) during stretching can be represented as (deduction

V_{\square} detail is shown in Note. S1):

$$P_t = \frac{V_{c,t} - V_{c,0}}{V_{\square} - V_{c,0}}$$

It can be observed that the maximum voltage limit V determine the value of the formula. Theoretically, V can take any arbitrary value. The reason for setting as 5 V in the text is to match with Arduino's serial port voltage levels.

According to your thoughtful comment, detailed explanations are added and revised on Page 9 (second paragraph) and Note S1 (Supplementary Information) of the revised manuscript in red font.

Amended in:

Page 9 (second paragraph): “This module transformed the initial resistance into a reciprocal voltage output (Fig. 3g), ensuring that the resulting output voltage was constrained within the range of 0-5 V. By maintaining a fixed voltage interval, stretch and touch variations could be decoupled via proportional scaling. The output voltage exhibited baseline-referenced bipolar variation, in which positive shifts corresponded to touch events, and negative shifts represented stretching. The final touch position was determined by the output voltage measured across the reference resistor, with the detailed calculation method provided in Note S1. This design facilitated efficient signal processing, while maintaining compatibility with the operating standards of portable measurement systems (limit of 5 V).”

Note S1 (Supplementary Information) “Temporal differential decoupling of stretch and touch position

The decoupling of the proprioceptive stretch signal and exteroceptive touch position signal is facilitated by the distinct patterns they exhibit. The stretch signal and touch position signal can each be modeled as linear changes in resistance. By characterizing the signal variations under specific stretching conditions, it becomes possible to accurately decouple and determine the relative touch position. The stretch can be quantified by comparing the voltage before ($V_{s,0}$) and after ($V_{s,t}$) stretching:

$$S_t = \frac{V_{s,t}}{V_{s,0}} \quad (1)$$

For a circuit composed of a constant resistor R_c and a stretchable resistor R_s , the voltage across the constant resistor is determined by the voltage divider rule:

$$V_c = V \frac{R_c}{R_c + R_s} \quad (2)$$

When no touch occurs, the readout voltage $V_r = V_c$ is influenced by the resistance values in the circuit. Since the constant resistor R_c is fixed, the voltage across it can be directly determined by the value of the stretchable resistor R_s . Thus, the touch voltage is given by:

$$V_{c,0} = V \frac{R_c}{R_c + R_{s,0}} \quad (3)$$

$$\frac{R_c}{R_c + R_{s,t}} = \frac{V_r}{V} \quad (4)$$

The relative touch position is defined as the ratio of the decrease in resistance due to the touch to the total resistance of the stretchable element:

$$P_t = \frac{R_{s,0} - R_{s,t}}{R_{s,0}} \quad (5)$$

Substituting R_s and $R_{s,t}$ into the position equation, we get:

$$P_t = \frac{V_c \cdot R_c}{V_c \cdot R_c + R_c} \left(\frac{V_c \cdot R_c}{V_c \cdot R_c + R_c} \right) \quad (6)$$

The final expression for the relative position of the touch point is:

$$P_t = \frac{V_c \cdot (V_{c,t} - V_{c,0})}{V_c \cdot (V_{c,t} - V_{c,0})} \quad (7)$$

Figure 3g:

Fig. 3g Voltage divider circuit diagram and corresponding signal patterns, demonstrating the conversion process from resistance to voltage.

5. In Fig. S12b, the word “Oberseved” may need further check.

Response: We are very grateful to you for carefully pointing out the typo in the figure. As you mentioned, the correct word should be “observed”. The typo is corrected in Fig. S13b (new number in the Supplementary Information).

Fig. S13. Simulated calculation of touch position under varying stretch factors.

Comments from Reviewer #2:

This paper presents a development of hyperconformal dual-modal metaskin for interactive hand motion interpretation by integrating a strongly coupled hydrophilic interface with a two-step transfer strategy to minimize interfacial mechanical losses.

This study is expected to provide a useful wearable sensor platform and this may be recommended for publication if the authors can successfully respond to the following comments.

Response: We sincerely appreciate your detailed review and very constructive feedback. Your comments are invaluable. We are fully dedicated to addressing them and making the necessary improvements to enhance the quality of our paper according to your great comments.

1. The authors said to ‘integrate both perceptual modes within the temporal framework, two key challenges must be addressed: achieving high-precision differentiation between them and capturing their coupling relationship in continuous motion. To better describe this relationship, a concept named “bodily context” is introduced, focusing on context estimation of motions besides the discrete definition. The concept of ‘bodily context’ is still not easy to understand and it had better be further explained.

Response: Thank you so much for highlighting these important aspects. In this work, the concept of bodily context is introduced to illustrate the approach of unifying proprioception and exteroceptive touch perception. Within this paradigm, hand motions and touch states are not only precisely decoupled but also labeled as bodily contexts. Specifically, when same tactile stimuli are applied to the metaskin under different hand flexion states, they generate distinct signal patterns—yet these signals cannot be directly interpreted. Our decoupling method separates these multimodal signals by assigning different body contexts to respective hand flexion states. Consequently, identical touch actions can be mapped to divergent interaction intentions depending on their contextual body states, thereby enabling comprehensive kinematic chain characterization. For instance, a "grasping posture" constitutes one body context, where identical wrist motions may represent different action chains when occurring in varying body contexts.

According to your helpful and to explain this conception, we add the detailed statements on Page 2 (first paragraph and second paragraph) of the revised manuscript in red font.

Amended in:

Page 2 (first paragraph): “To temporally integrate both perceptual modes, a concept named “bodily context” is introduced, which associates proprioceptive hand states with touch patterns. This approach not only decouples touch stimuli from hand motions but also contextualizes them: identical tactile inputs yield distinct perceptual results under different postures, and vice versa. By resolving these multimodal relationships, the same touch action can express divergent intentions on the basis of its context, enabling natural motion-chain interpretation. The motion coupling based on the bodily context will revolutionize the interactive model, enabling cognition and judgement depending on the context in which it occurs.”

Page 2 (second paragraph): “However, touch information is often overlooked in gesture recognition frameworks, resulting in systems that still deviate from the natural kinematic chain of hand actions. By integrating contextual awareness with motion cognition, the hand’s proprioceptive state, i.e., the bodily context, directly influences touch perception, and vice versa. This integration breaks the traditional occlusion between proprioceptive motion and tactile modalities.”

2. Due to the strong adhesion between the nano-composite paste and the WPU substrate, the HDM metaskin transferred to the skin exhibits exceptional electrical and mechanical robustness. It maintains stable electrical resistance under prolonged frictional wear, remains unaffected by high humidity (90% RH), and retains thermal stability across a wide temperature range. These attributes ensure reliable performance in diverse and dynamic environments (Fig. 2e). Usually, metal shows a PTC behavior where the resistance increases as the temperature increase. I wonder why the metaskin does not show temperature dependence.

Response: We deeply appreciate your careful review and insightful suggestions. The temperature coefficient of resistance (TCR) for pure metals typically ranges from $+0.003/^{\circ}\text{C}$ to $+0.004/^{\circ}\text{C}$. Compared to semiconductors and functional materials, metals exhibit weak positive temperature coefficient (PTC) effects. When considering metallic PTC behavior, the resistance variation remains around 10% (25-60 $^{\circ}\text{C}$), which has a little practical impact.

To further address the concern, we conduct impedance measurements under continuous heating. The results demonstrate that nano-Ag filled composites show merely $\sim 4\%$ impedance variation within the safe touch temperature range (30-60 $^{\circ}\text{C}$) with minimal changes at 100 $^{\circ}\text{C}$. In comparison, micro-Ag filled composites exhibit less stable variations during heating, though overall impedance fluctuations remain controllable.

These observations indicate that: (1) The composite system mitigates metallic PTC effects, likely due to discontinuous conductive pathways; (2) Size effects exist. Nanoparticles form denser conductive networks that are less sensitive to matrix expansion and particle rearrangement during heating.

According to your insightful suggestions, we add the detailed explanation in Fig. S9 (Supplementary Information) and on Page 7 (first paragraph) of the revised manuscript in red font.

Amended in:

Page 7 (first paragraph): “It maintained stable electrical resistance under prolonged frictional wear (30 m, 2 N, 10 cm s⁻¹) and high humidity (90% RH), while resisting resistance changes under thermal conditions. Even across the safe temperature range (25–60°C), the resistance variation remained below 5% (Fig. 2e and Fig. S9). These attributes ensure reliable performance in diverse and dynamic environments.”

Added Figure:

Fig. S9 Impedance stability comparison in the 30–100°C range for (a) nano-Ag and (b) micro-Ag filled composites.

Thermal stability of Ag-filled conductive composites was investigated through continuous heating test. Both nano- and micro-Ag composites exhibit significantly suppressed positive temperature coefficient (PTC) effects compared to pure metals due to their discontinuous conductive medium. While the nano-Ag composite demonstrated exceptional stability with negligible impedance variation (about 4% in the safe temperature range from 30°C to 60°C), the micro-Ag composites showed greater temperature sensitivity, displaying measurable fluctuations even below 60°C. The superior

performance of nano-Ag composites suggests their densely-packed networks are more resistant to potential thermal expansion and particle rearrangement.

Figure 2e:

Fig. 2e Robust performance of the HDM metaskin across friction, humidity conditions (90% RH), and temperature ranges from 25 to 60°C.

3. Figure 5 shows mapping of proprioceptive and exteroceptive movements to five different robot commands using a U-shaped meta-skin installed on the wrist. The system interpreted the touch positions to navigate the walking directions of the hexapod robot, and the wrist bending states mapped to the walking postures—wrist flexion causing lifting of the machine body. The robust decoupling of bi-modal signals allowed us to relocate the command index position even in different wrist bending states. The walking profile of robot correlated to hand bending states, and under different command contexts, the robot would move with an appropriate stride to align with the real environment. It seems to me that the robot can do a specific job depending on the location of touch. I wonder if the command can be more sophisticated (for example, the move distance, angle, speed etc).

Response: We highly appreciate your helpful comment. To demonstrate precise control capabilities, we enhance the original single-action command control logic by integrating temporal control parameters to regulate movement distance. Furthermore, the system now attains precise control of move speed and angle based on variations in touch position.

According to your kind suggestion, these improvements are documented in the new Movie S3 and Fig. S19 (Supplementary Information) and the corresponding descriptions are added on Page 12 (first paragraph) of the revised manuscript in red font.

Amended in:

Page 12 (first paragraph): “In another example, precise navigation of the hexapod robot was demonstrated using a linear meta-skin, highlighting the expanded capabilities in both form and

function of meta-skins (Fig. S19 and Movie S3).”

Added Figure:

Fig. S19 Hexapod robot precise navigation with a line-type HDM metaskin.

(a) Morphology of the line-type HDM metaskin on the wrist (scale bar: 5 cm). (b) Control setup for the metaskin. The HDM metaskin allows different control modes depending on its stretching. In its original state, it functions as a moving context, where the touch position indicates the movement direction and speed. When stretched, it switches to a turning context, allowing precise steering angle adjustments. (c-f) Real-world scenes showing the hexapod robot's navigation, demonstrating accurate control of distance and angles.

4. The metaskin leverages the integration of the ultrathin hydrophilic substrate (8–10 μm) and highly compatible water-based Ag nanocomposite sensitive path, minimizing impediments on hand movements. However, 8-10 μm film had better be called thin substrate. Ultrathin e-skin usually refers to the sub-micron thin film substrate.

Response: Thanks very much for your nice suggestion. According to your suggestion, we remove all the terms “ultrathin” from the manuscript and replace them with “thin-film” and “thin”.

Amended in:

Page 3 (last paragraph): “The metaskin leverages the integration of the hydrophilic thin substrate (8–10 μm) and highly compatible water-based Ag nanocomposite sensitive path, minimizing impediments to hand movements.”

Page 5 (last paragraph): “Because the thin-film substrate was susceptible to distortion without a support film and because the conductive paths needed to be the outer surface”

Page 6 (third paragraph): “Next, we focused on interface engineering to elucidate the issues involved in the formation and transfer of thin films. Owing to the hydrophilic nature of WPU, thin films cannot be successfully spin-coated from dilute solutions on hydrophobic substrates. [...] Mechanical testing demonstrated that the thin-film substrate exhibited negligible tensile forces on the skin, minimizing any foreign body sensation.”

Page 16 (first paragraph): “The strong interfacial coupling between the thin hydrophilic substrate and the water-based Ag nanocomposite is the physical basis of the mechanoreceptor”

Page 17 (second paragraph): “HDM metaskin was achieved through the fabrication of thin films and the printing of sensitive Ag paths, followed by a secondary transfer process.”

Page 17 (third paragraph): “A thin layer of pressure-sensitive adhesive (PSA) was then spin-coated onto the exposed WPU surface and subsequently dried on a hot plate at 50°C for 10 minutes.”

Page 18 (second paragraph): “The strain and stress distributions of the thick and thin- film electronics were compared through the finite element method (version 5.6, COMSOL Multiphysics). [...] and the thicknesses of the thick film and thin film were set as 40 μm and 10 μm , respectively.”

5. From the video clips for the hexapod robot, typing, mouse control and drumming, the response

seems to be a bit delayed. What is the limiting factor for delay and how can it be improved?

Response: We deeply appreciate your careful observation and insightful suggestions.

The hexapod robot employed in our study possesses limited capabilities—its movement and lifting functions depend on predefined motion sets rather than real-time twin control. Moreover, to enhance the clarity of the demonstration, the robot was configured to respond solely after the confirmation of hand movement completion. These issues may be addressed by refining the control logic or upgrading to a higher-performance robotic system. In response to the reviewer's concerns, we implement real-time command streaming in a new demo (Movie S3), replacing single-action commands, thereby effectively eliminating the visual delay.

The latency associated with typing and mouse control is comparatively minimal. The visual delay primarily originates from the program logic. These concerns may be mitigated through the optimization of program logic and enhancements in multithreading techniques.

Regarding the drumming demonstration, the sound produced after drumming is generated by playing a pre-edited audio file. The primary causes of the perceived delay are the loading time of the audio file and the inherent silent segments within the audio itself. Moreover, action recognition employs a fixed time window in our machine learning model, with modal activation conditioned upon completed wrist motion verification. Visual latency in the drumming demonstration arises from media file loading delays and latency during instruction processing. These issues can be resolved by optimizing the file loading pipeline and implementing data window preloading strategies.

According to your comment, we analyze the main limiting factors of latency in different demo applications and provide corresponding optimization methods on Page 19 (second and third paragraph) of the revised manuscript in red font.

Amended in:

Page 19 (second paragraph): “In Movie S2, to improve demonstration clarity, the robot responded only after full confirmation of hand movement completion. This configuration introduced perceptible visual latency, particularly during complex motion sequences. As addressed in Movie S3, switching from single-action commands to real-time streaming eliminated such delays. Refining control logic or upgrading hardware may further optimize system responsiveness.”

Page 19 (third paragraph): “Context-switching drumming demonstration

The drumming demonstration was implemented through synergistic control of the HDM metaskin on index finger and wrist. The drumstick was attached to conductive tracings to trigger external contact events. For the demonstration procedure, we implemented pre-edited audio file playback triggered by drumming actions to generate sound feedback, and implemented image playback to generate visual feedback. Action recognition employs a fixed time window in our machine learning model, where the modal activation is triggered only after the system confirms wrist motion completion. Visual latency in the drumming demonstration arises from media file loading delays and latency during instruction processing. These issues can be resolved by optimizing the file loading pipeline and implementing data window preloading strategies.”

6. Various soft e-skin sensors for hand motion recognition have been developed recently. Very recently, nanomaterial based soft sensors with machine learning for hand motion detection (Nat. Electron., 6, 64, (2023); Nat. Comm., 11, 2149, (2020)) have been demonstrated for virtual keyboard/touch and shape recognition. They should be briefly introduced in the manuscript and the novelty compared with those recent studies needs to be discussed.

Response: We highly appreciate your nice recommendation. The recommended studies align with the scope on hand motion recognition and imperceptible wearable devices, particularly in demonstrating cutaneous e-skins for capturing minute hand motions and skin deformations. Overall, those studies are very interesting. Existing e-skins generate single-modal haptic feedback, which deviates from the natural kinematic chain of hand actions. The novelty of our HDM metaskin lies in its dual-modal integration of proprioceptive skin stretches and exteroceptive touch signals, grounded in a somatic motion interpretation framework. It enables precise intention recognition and task adaptation by leveraging the complementary relationship between tactile and proprioceptive information.

According to your constructive feedback, we amend the discussion of this significance on Page 2 (third paragraph) and Page 3 (second paragraph) of the revised manuscript in red font.

Amended in:

Page 2 (third paragraph): “Thus, imperceptible soft e-skins that capture high-precision contact information are gaining attention, driven by a strong desire to produce devices that can be integrated into everyday actions without constraining the users¹⁶. Recent advances in the in-situ printing of

nanomesh e-skin have achieved biomimetic sensing and imperceptible implementation, directly mapping microscale skin stretches to proprioception¹⁷. Furthermore, to truly replicate biological tactile intelligence, where actions like grasping rely on a strong correlation between touch and muscle kinetics, decoding proprioception–touch interdependence becomes fundamental. This helps to offer new insight into the adjustment of interaction feedback and manipulation mode switching on the basis of the bodily context.”

Page 3 (second paragraph): “It captures subtle resistance changes from skin stretching and electrical contact localization, precisely decoding touch positions even under stretching. The core mechanism leverages polarity differences, where stretch signals increase resistance while touch events decrease it proportionally. By interpreting hand motions as combinations of bodily context and event–action signals, the seamless integration of proprioceptive and exteroceptive data enhances context–kinematics coupling, enabling precise intention recognition and task adaptation.”

Added References:

[16] Kim, K. K. *et al.* A deep-learned skin sensor decoding the epicentral human motions. *Nat. Commun.* **11**, 2149 (2020).

[17] Kim, K. K. *et al.* A substrate-less nanomesh receptor with meta-learning for rapid hand task recognition. *Nat. Electron.* **6**, 64 (2023).

7. Hand gesture plays a very crucial role in human-machine interface and VR interface. Hand gesture recognition for interfacing with VR application (*Adv. Funct. Mater.*, 31, 2007772 (2021); *Adv. Funct. Mater.*, 32, 2106329, (2022); *Natl. Sci. Rev.*, 11, nwad298 (2024)) needs to be briefly discussed in the introduction part to help the readers with the recent related progresses.

Response: Your insightful recommendation to consult the cited works significantly strengthened our perspective. Gesture recognition is highly relevant to our research focus and constitutes a critical component of this study.

According to your constructive feedback, we expand the discussion of its significance on Page 2 (second paragraph) of the revised manuscript in red font.

Amended in:

Page 2 (second paragraph): “Wearable devices capture and interpret hand motion and touch to

convey the interactive intentions of humans, such as object manipulation or gesture recognition. Gesture recognition complements perception beyond vision, playing an indispensable role in immersive virtual reality and communication assistance. Recent advances in markerless gesture measurement have aimed to achieve a natural description of hand movements¹²⁻¹⁴.”

Added References:

[12] Oh, J. *et al.* A liquid metal based multimodal sensor and haptic feedback device for thermal and tactile sensation generation in virtual reality. *Adv. Funct. Mater.* **31**, 2007772 (2021).

[13] Kim, K. K., Choi, J., Kim, J., Nam, S. & Ko, S. H. Evolvable skin electronics by in situ and in operando adaptation. *Adv. Funct. Mater.* **32**, 2106329 (2022).

[14] Pyun, K. R. *et al.* Machine-learned wearable sensors for real-time hand-motion recognition: toward practical applications. *Natl. Sci. Rev.* **11**, nwad298 (2024).

8. A U-shaped HDM metaskin was affixed to the user’s wrist, with a wireless module connected to its terminal. Single-channel data comprising somatic action information was transmitted to a personal computer for further differentiation processing. A software interface developed in Python collected these data and managed the conversion and transmission of control signals. The communication system utilized a pair of HC-05 Bluetooth modules configured in master-slave mode to establish a reliable wireless connection between the PC and the hexapod robot. I wonder why U-shaped configuration was used for this application. Does it require long length?

Response: We sincerely appreciate you for highlighting this significant point. It is beneficial for conforming to the natural curvature of the wrist for robot navigation applications. We aim to demonstrate the shape customisation capability of the metaskin by designing tailored configurations for different body parts. For the wrist, we select a U-shaped configuration to increase the number of interaction points and improve visual demonstration effectiveness.

According to your helpful comment and to clarify this point, a shorter line-type sensing path (7 cm) is demonstrated in the new Movie S3 and Fig. S19 (Supplementary Information). The detailed explanation is added and presented on Page 11 (second paragraph) of the revised manuscript in red font.

Amended in:

Page 11 (second paragraph): “For the wrist, we choose a U-shaped configuration to both increase the number of interaction points and enhance the visual demonstration effectiveness.”

Page 11 (second paragraph): “In another example, precise navigation of the hexapod robot was demonstrated using a linear meta-skin, highlighting the expanded capabilities in both form and function of meta-skins (Fig. S19 and Movie S3).”

Added Figure:

Fig. S19 Hexapod robot precise navigation with a line-type HDM metaskin.

(a) Morphology of the line-type HDM metaskin on the wrist (scale bar: 5 cm). (b) Control setup for the metaskin. The HDM metaskin allows different control modes depending on its stretching. In its original state, it functions as a moving context, where the touch position indicates the movement direction and speed. When stretched, it switches to a turning context, allowing precise steering angle adjustments.

9. The general usage of language including typos and grammatical errors need to be checked again.

Response: We highly appreciate your kind reminder. We carefully re-read and conduct a thorough spell check to ensure the clarity and accuracy of the manuscript. We also utilize Springer Nature Editing Service to further refine the language. The certificate is as follows. Thanks so much once again for your great suggestion and help.

This document certifies that the manuscript
A hyperconformal dual-modal metaskin for well-defined and high-precision
contextual interactions

prepared by the authors

Shifan Yu

was edited for proper English language, grammar, punctuation, spelling, and overall style
by one or more of the highly qualified English speaking editors at SNAS.

This certificate was issued on **July 10, 2025** and may be verified
on the SNAS website using the verification code **7792-7F91-E8E9-E5C7-5D41**.

Neither the research content nor the authors' intentions were altered in any way during the editing process. Documents receiving this certification should be English-ready for publication; however, the author has the ability to accept or reject our suggestions and changes. To verify the final SNAS edited version, please visit our verification page at secure.authorservices.springernature.com/certificate/verify.
If you have any questions or concerns about this edited document, please contact SNAS at support@es.springernature.com.

SNAS provides a range of editing, translation, and manuscript services for researchers and publishers around the world.
For more information about our company, services, and partner discounts, please visit authorservices.springernature.com.

Amended in:

Page 1 (Abstract): “The 10- μm -scale hyperconformal film is highly sensitive to intricate skin stretches while minimizing signal distortion. It accurately tracks skin stretches as well as touch locations and translates them into polar signals, which are individually salient. This approach enables a differentiable signalling topology within one single data channel without burdening structural complexity to the metaskin. When combined with temporal differential calculations and time-series machine learning network, the metaskin extracts interactive context and action cues from the low-dimensional data. This phenomenon is further exemplified through demonstrations in contextual navigation, typing and control integration, and multi-scenario object interaction. We demonstrate this fundamental approach in advanced skin-integrated electronics, highlighting its potential for instinctive interaction paradigms and paving the way for augmented somatosensation recognition.”

Page 6 (third paragraph): “Next, we focused on interface engineering to elucidate the issues involved in the formation and transfer of thin films. Owing to the hydrophilic nature of WPU, thin films cannot be successfully spin-coated from dilute solutions on hydrophobic substrates. As shown in Fig. 2c, when a 30 vol% WPU solution was spin-coated at $1000 \text{ r}\cdot\text{min}^{-1}$, the liquid underwent

annular contraction driven by the imbalance of Young–Laplace forces. This phenomenon occurred because dilute solutions tend to exhibit higher surface tension due to water aggregation, coupled with low viscous resistance, which promotes liquid redistribution.”

Page 6 (last paragraph): “The transfer process involves the interaction and migration of four key interfaces: the release paper, the WPU, the silicone transfer film, and the target interface. In this context, each transfer inherently replaces a low-interaction interface with a high-interaction interface, placing additional demands on the transfer auxiliary film to establish a transient adsorption state. ... Notably, this outcome is not merely a result of surface energy ranking, even though the silicone film typically has a lower surface energy than does the release paper. Owing to the strong adhesion between the nanocomposite paste and the WPU substrate, the transfer of HDM metaskin transferred to the skin exhibits exceptional electrical and mechanical robustness. It maintained stable electrical resistance under prolonged frictional wear (30 m, 2 N, 10 cm s⁻¹) and high humidity (90% RH), while resisting resistance changes under thermal conditions. Even across the safe temperature range (2560°C), the resistance variation remained below 5% (Fig. 2e and Fig. S9). These attributes ensure reliable performance in diverse and dynamic environments.”

Page 8 (last paragraph): “The consistency of the signal at the same contact position was also ensured (Fig. 3d). In contrast, the stretch-induced resistance signal was positive and initiated from the base value. Owing to the interlocking network and interpenetrating interfaces, the Ag nanocomposite path exhibited a highly sensitive response, with a gauge factor of 32.45 in the strain range of 0 to 20%, and demonstrated enhanced sensitivity up to a strain of 50% (Fig. 3e). Additionally, we compared the cyclic resistance recoveries of Ag NPs, Ag NPs/carbon nanotubes (CNTs), and Ag NPs/Ag NWs to highlight the crucial role of the composite network (Fig. 3f). Without the anchoring effect of Ag NW, the Ag NP path experienced pronounced signal fluctuations and drift under a cyclic strain of 5%. This was probably attributed to the unstable expansion and delayed recovery of the nanocracks. Introducing CNTs partially reduced these fluctuations, yet minor variations remained. In contrast, incorporating Ag NWs resulted in a more robust conductive network, yielding consistently lower and more stable resistance changes.”

Page 9 (second paragraph): “This module transformed the initial resistance into a reciprocal voltage output (Fig. 3g), ensuring that the resulting output voltage was constrained within the range of 0-5 V. By maintaining a fixed voltage interval, stretch and touch variations could be decoupled via

proportional scaling. The output voltage exhibited baseline-referenced bipolar variation, in which positive shifts corresponded to touch events, and negative shifts represented stretching. The final touch position was determined by the output voltage measured across the reference resistor, with the detailed calculation method provided in Note S1. This design facilitated efficient signal processing, while maintaining compatibility with the operating standards of portable measurement systems (limit of 5 V).”

Page 9 (last paragraph): “By utilizing limited information from one-dimensional signals, it decouples bimodal signals while simultaneously accounting for both signal characteristics and temporal dynamics. Since this approach requires temporal sequence analysis rather than single-time-point signals, it does not support static motion benchmarking. Therefore, this study focused on interpreting hand motions within defined time windows, as it provided the possibility of coupling skin stretch and touch motions in continuous temporal framework.”

Page 11 (last paragraph): “These merging signals were then processed by filtering and encoding, providing dynamic and contextual cues for task refinement (Fig. 5b). For example, a motor robot must adjust its moving posture and moving speed under variable terrain conditions. As shown in Fig. 5c, the system interpreted the touch positions to navigate the walking directions of a hexapod robot, and the wrist bending states mapped to the walking postures—wrist flexion causing lifting of the machine body. The robust decoupling of bimodal signals allowed for us to relocate the command index position even in different wrist bending states. By harnessing the information abundance in a single device, we elaborated the adaptive interaction in a live navigation task (Fig. 5d and Movie S2). The walking profile of the robot is correlated with the hand bending state, and under different command contexts, the robot moves with an appropriate stride to align with the real environment. In another example, precise navigation of the hexapod robot was demonstrated using a linear meta-skin, highlighting the expanded capabilities in both form and function of meta-skins (Fig. S19 and Movie S3). As shown previously, the decoupling calculation of the touch position was not affected by the stretching state, allowing for accurate positioning of the relative touch position even when the context changed. Therefore, the HDM metaskin exhibited an extended ability to reactively and anticipatorily recognize the interactive intentions of humans, as naturally as a skin extension.

Page 15 (second paragraph): “This confusion arose because wrist movement signals do not noticeably change with different grip postures, making accurate extraction of static features using

machine learning methods nearly impossible. Thus, to coordinate the matching of the interaction posture and dynamics, multidimensional information from both the grip posture and wrist movements must be considered simultaneously.”

Page 18 (second paragraph): “The strain and stress distributions of the thick and thin- film electronics were compared through the finite element method (version 5.6, COMSOL Multiphysics). The width and depth of the skin wrinkles were set as 160 μm and 80 μm , respectively. A hyperelastic WPU material model was applied above the wrinkle, and the thicknesses of the thick film and thin film were set as 40 μm and 10 μm , respectively. The Ag composite conductive path was set at a thickness 10 μm . The strain curve was obtained by extracting the strain data from the top surface of the Ag conductive path.”

10. Some of the digital pictures are missing scale bars. They should be added to the pictures.

Response: Thanks very much for your kind suggestion. According to the suggestion, we add scale bars to Fig. 2d, Fig. 4f, Fig. 5c, Fig.5d, Fig. 5f, Fig. 5h, and Fig. 6f, and annotate the length in the figure captions.

Revised Figures:

Fig. 2d Scale bar: 2 cm.

Fig. 4f Scale bar: 2 cm.

Fig. 5c Scale bar: 5 cm.

Fig. 5d Scale bar: 10 cm.

Bi-finger typing

Keyboard input

Fig. 5f Scale bar: 2 cm.

Fig. 5h Scale bar: 2 cm.

Fig. 6f Scale bar: 1 cm.

We sincerely appreciate Dr. Ellasia Tan and Reviewers' encouraging evaluation and constructive comments, and try our best to improve the manuscript. Hope the improved paper will be satisfied for the publication in *Nature Communications*. Thank you with the utmost respect for your consideration.

With kind regards,

Zhong Chen

Ph.D., Minjiang Distinguished Professor, Dean
School of Electronic Science and Engineering
Xiamen University
Xiamen 361005, China
[E-mail: chenz@xmu.edu.cn](mailto:chenz@xmu.edu.cn)

Yuanjin Zheng

Ph.D., Professor, Department Chairman
School of Electrical and Electronic Engineering
Nanyang Technological University
Singapore 639798, Singapore
[E-mail: yjzheng@ntu.edu.sg](mailto:yjzheng@ntu.edu.sg)

Qingliang Liao

Ph.D., Changjiang Distinguished Professor,
Dean School of Materials Science and
Engineering University of Science and
Technology Beijing Beijing 100083, China
[E-mail: liao@ustb.edu.cn](mailto:liao@ustb.edu.cn)

Xinqin Liao

Ph.D., Associate Professor
School of Electronic Science and Engineering
Xiamen University
Xiamen 361005, China
[E-mail: liaoxinqin@xmu.edu.cn](mailto:liaoqxinqin@xmu.edu.cn)

Response Letter (Manuscript ID: NCOMMS-25-31344A)

Dear Reviewers,

We are very grateful for your efforts and time in processing our manuscript (**Manuscript ID: NCOMMS-25-31344A**). Below, we list all comments from the Reviewers and provide our point-by-point responses (in **blue font**).

Comments from Reviewer #1:

In this version, the authors have provided additional experiments to address the concern, including experimental results and the supporting of characterization system. The concerns have been clarified, and the discussion is well organized to support the claims in the manuscript.

I would thus recommend acceptance of the current work in *Nature Communications*.

Response: We sincerely appreciate your positive feedback on our revised manuscript. Your positive evaluation and recommendation for publication are greatly encouraging.

Comments from Reviewer #2:

The authors responded well to the comments.

Response: Thank you very much for your thoughtful and constructive feedback. We sincerely appreciate your recognition.